# MULTI-HEAD STATE SPACE MODEL FOR SEQUENCE MODELING

## ABSTRACT

Recently, state space models (SSMs) have shown promising results on sequence modeling tasks. However, a potential challenge of existing works is that SSMs are usually introduced or initialized in a homogeneous way, encouraging the model to only capture similar temporal dynamics on different features. In this paper, we propose a multi-head state space model (MSSM), in which parallel heads are introduced to learn different temporal dynamics on sequence data. Furthermore, we propose a novel variant of the Transformer, referred to as the Stateformer, which combines MSSMs with attention. Experiments on large-scale automatic speech recognition (ASR) and language modeling tasks show the MSSM outperforming a range of attention-based baselines. The Stateformer further improves performance, achieving the state-of-the-art performance on the LibriSpeech ASR task.

## 1 INTRODUCTION

Transformers (Vaswani et al., 2017) and attention based models (Bahdanau et al., 2015) have for more than half a decade shown state-of-the-art performance on a wide range of tasks anywhere from speech recognition (Zhang et al., 2020a; Gulati et al., 2020) and neural machine translation (Ng et al., 2019; Chen et al., 2020; Tran et al., 2021) to computer vision (Dosovitskiy et al., 2021; Liu et al., 2021b) and biological applications such as protein sequence modeling (Jumper et al., 2021; Rives et al., 2021). One of the strengths of these models comes from having a significantly smaller maximum path lengths across time, which is the shortest path linking the first encoder input and final decoder output (Hochreiter et al., 2001). This is unlike previous state-of-the-art recurrent and convolutional neural networks which have linearly and logarithmically scaling path lengths respectively. However, a well known drawback of transformers is the quadratic space and time complexity of the self-attention layer, restricting its applicability to fields requiring longer sequences or to devices with strict limitations on compute resources (Tay et al., 2022). To combat both of these issues, a wide range of restricted sparse attention mechanisms have been proposed, all aiming at both reducing the computational cost and scaling to longer sequences but retaining as much of the original transformer performance as possible (Katharopoulos et al., 2020; Kitaev et al., 2020; Beltagy et al., 2020; Zaheer et al., 2020). There has also been a notable amount effort been poured into designing alternative more efficient attention schemes instead of using the dot-product approach (Wang et al., 2020a; Choromanski et al., 2020; Shen et al., 2021; Zhang et al., 2019).

Meanwhile, the machine learning community is paying more attention to a well established signal processing and control theory technique, the state space model (SSM) (Kalman, 1960), which historically has been widely used in many time-series and control problems (Brogan, 1991; Hyndman et al., 2008; Durbin & Koopman, 2012). More recently, the linear time-variant state space model has also been used within neural networks for improving time-series forecasting (Seeger et al., 2017; 2016; Rangapuram et al., 2018). Furthermore, a simplified linear time-invariant (LTI) SSM, which is closely related to fully linear recurrent neural network (RNN) layers, has a well-known convolution equivalent making it particularly attractive for parallelized training while inference mode can utilize its fast recurrent formulation (Brogan, 1991; Gu et al., 2021).

One potential limitation of existing works in this direction is that SSMs are usually initialized and used in a homogeneous way. For instance, in Gu et al. (2022a), the S4 approach equips SSMs with careful options on parameter initialization for long-range modeling ability. This design would force the model to capture similar temporal dynamics on different features. As long and short term

dependency would both be useful in sequence modeling, in this paper, we develop a multi-head state space model, which consists of parallel heads to learn different temporal dynamics on sequence data. We investigate the use of multi-head state space models in sequence tasks as both a replacement and complement to attention. Since SSMs have been shown to be promising on long sequences, we hypothesize such a model is able to handle both short and long-term dependencies and can operate as an effective alternative to attention. Following are our main technical contributions around the state space model:

1. *Stacked and multi-head generalization.* We extend the SSM approach by allowing multi-head parallel processing of projected lower-dimensional signals and stack such a layer (within a residual block) for better performance.

2. *Head gating.* We propose an inter-head gating approach allowing different SSMs within the multi-head layer to communicate.

3. *Combination with attention.* We also augment the transformer architecture by simply including a bidirectional SSM residual block prior to the attention for computationally unrestricted applications and state-of-the-art performance; referred to as the *Stateformer*.

With these contributions, we advance the state of attention-free models on large-scale speech recognition and language modeling tasks, and show that our novel multi-head state space model outperforms strong attention-based baselines. We also show that combining the multi-head state space with attention can achieve state-of-the-art performance in large-scale speech recognition.

## 2 BACKGROUND

This section will cover theory and related work regarding the state space model and its use in deep learning. We first start by briefly covering the state space model and a block diagonal structured restriction. We further mention a convolution point of view for parallel training (Section 2.1). We finish with a brief discussion of the state space layer as used in neural nets (Section 2.2).

### 2.1 STATE SPACE MODEL: THE LINEAR RNN

A specific realization of the state space model (SSM) is a linear time-invariant (LTI) model (Brogan, 1991) that transforms some input signal $\boldsymbol{u}(t) \in \mathbb{R}^{D_i}$ to some output $\boldsymbol{y}(t) \in \mathbb{R}^{CD_i}$ through some hidden process $\boldsymbol{x}(t) \in \mathbb{R}^{D_h}$ according to

$$
\begin{aligned}
\dot{\boldsymbol{x}}(t) &= \boldsymbol{A}\boldsymbol{x}(t) + \boldsymbol{B}\boldsymbol{u}(t), \\
\boldsymbol{y}(t) &= \boldsymbol{C}\boldsymbol{x}(t) + \boldsymbol{D}\boldsymbol{u}(t)
\end{aligned}
\tag{1}
$$

where $\boldsymbol{D}\boldsymbol{u}(t)$ can be neglected since this is a simple term to compute. To ensure such a model is compatible with discrete signals such as sequences of speech frames or sub-word unit embeddings, a simple discretization, e.g. the bilinear method (Tustin, 1947) can be utilized:

$$
\begin{aligned}
\boldsymbol{x}_k &= \overline{\boldsymbol{A}}\boldsymbol{x}_{k-1} + \overline{\boldsymbol{B}}\boldsymbol{u}_k, & \overline{\boldsymbol{A}} &= (\boldsymbol{I} - \boldsymbol{\Delta}\boldsymbol{A}/2)^{-1}(\boldsymbol{I} + \boldsymbol{\Delta}\boldsymbol{A}/2), \\
\boldsymbol{y}_k &= \overline{\boldsymbol{C}}\boldsymbol{x}_k, & \overline{\boldsymbol{B}} &= (\boldsymbol{I} - \boldsymbol{\Delta}\boldsymbol{A}/2)^{-1}\boldsymbol{\Delta}\boldsymbol{B}, \ \overline{\boldsymbol{C}} = \boldsymbol{C}
\end{aligned}
\tag{2}
$$

which simply represents a fully linear recurrent neural network (RNN) cell. Note that the discretization matrix $\boldsymbol{\Delta} \in \mathbb{R}^{D_h \times D_h}$ is diagonal with positive entries, but can be subsumed by $\boldsymbol{A}, \boldsymbol{B}$ due to scale invariance. Furthermore, while such a model is highly flexible, providing it with structure can increase its efficiency. Therefore, one can restrict the matrices $\boldsymbol{A}, \boldsymbol{B}, \boldsymbol{C}$ to have a block diagonal structure as follows:

$$
\boldsymbol{A} = \begin{pmatrix} \boldsymbol{A}_1 & 0 & \dots & 0 \\ 0 & \boldsymbol{A}_2 & \dots & 0 \\ \vdots & \vdots & \ddots & \vdots \\ 0 & 0 & \dots & \boldsymbol{A}_{D_i} \end{pmatrix}, \ \boldsymbol{B} = \begin{pmatrix} \boldsymbol{B}_1 & 0 & \dots & 0 \\ 0 & \boldsymbol{B}_2 & \dots & 0 \\ \vdots & \vdots & \ddots & \vdots \\ 0 & 0 & \dots & \boldsymbol{B}_{D_i} \end{pmatrix}, \ \boldsymbol{C} = \begin{pmatrix} \boldsymbol{C}_1 & 0 & \dots & 0 \\ 0 & \boldsymbol{C}_2 & \dots & 0 \\ \vdots & \vdots & \ddots & \vdots \\ 0 & 0 & \dots & \boldsymbol{C}_{D_i} \end{pmatrix}
\tag{3}
$$

where the structure of this block recurrent model is based on the number of input dimensions $D_i$[1]. The original multi-variate state space model can now be seen as $D_i$ smaller independent sub-SSMs, where each one of these smaller models are $\mathbb{R} \to \mathbb{R}^C$ mappings running in parallel.

---

[1]A block diagonal treatment of structured state spaces is also covered in Smith et al. (2022)

Denoting the sub-hidden dimension by $D_{sh} = D_h/D_i$ we obtain $\boldsymbol{A}_n \in \mathbb{R}^{D_{sh} \times D_{sh}}$, $\boldsymbol{B}_n \in \mathbb{R}^{D_{sh} \times 1}$ and $\boldsymbol{C}_n \in \mathbb{R}^{C \times D_{sh}}$. The skip-connection can also be split into blocks of $\boldsymbol{D}_n \in \mathbb{R}^{C \times 1}$. Similarly, the discretization matrix can now be seen as a diagonal matrix with a isotropic blocks $\boldsymbol{\Delta}_n = \Delta_n \boldsymbol{I}$ of the same size as the transition blocks $\boldsymbol{A}_n$, meaning that each sub-state space model has its own discretization time-step that can be tuned to its own input channel.

Furthermore, due to the model being fully linear, one can utilize the widely known duality between convolutions and recurrences, resulting in an efficient expression:

$$\overline{\boldsymbol{K}} = (\overline{\boldsymbol{CB}}, \overline{\boldsymbol{CAB}}, \dots, \overline{\boldsymbol{CA}}^L \overline{\boldsymbol{B}}), \quad \boldsymbol{y} = \overline{\boldsymbol{K}} * \boldsymbol{u} \tag{4}$$

where $L$ is the input sequence length. Therefore, state space models and linear RNNs can be trained in convolution mode (Equation 4), fully utilizing the power of GPUs, and be deployed using recurrent mode (Equation 2), efficiently and incrementally generating predictions. To further improve the efficiency, a specific diagonal plus low-rank restriction of the sub-transition blocks can be used Gu et al. (2022a),

$$\boldsymbol{A}_n = \boldsymbol{\Lambda}_n + \boldsymbol{p}_n \boldsymbol{q}_n^T, \quad \boldsymbol{p}_n, \boldsymbol{q}_n \in \mathbb{R}^{D_{sh}} \quad \forall n = \{1, \dots, D_i\} \tag{5}$$

where $\boldsymbol{\Lambda}_n$ is a diagonal matrix. This allows the kernel $\overline{\boldsymbol{K}}$ to be computed efficiently utilizing the Woodbury matrix identity with several fast implementations (Pan, 2001; 2014).

## 2.2 PRACTICAL LAYER DESIGN

The structured state space model described in Equation 2 is a general linear $\mathbb{R}^{D_i} \to \mathbb{R}^{CD_i}$ mapping but processes each channel independently due to the structuring in Equation 3. To amend both these problems, a non-linear activation is introduced followed by a point-wise linear layer (see Figure 1a for a pre-norm residual unit) to mix the information across input channels:

$$\boldsymbol{f}(\boldsymbol{u}) = \texttt{Linear}(\texttt{Act}(\texttt{SSM}(\boldsymbol{u}))) \tag{6}$$

Note the output dimension expansion $C$ is set to 1 when using element-wise activations such as GELU (Hendrycks & Gimpel, 2016) or set to 2 when using a gating mechanisms such as GLU (Dauphin et al., 2017).

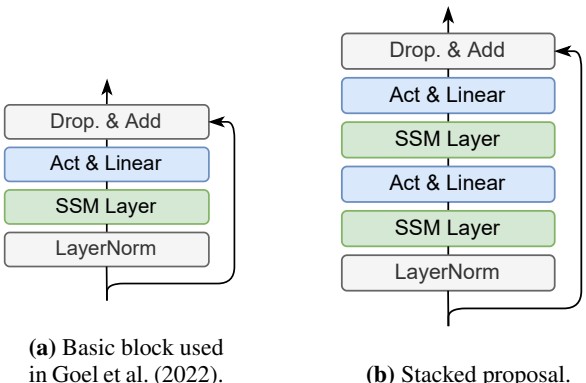

**(a)** Basic block used in Goel et al. (2022).

**(b)** Stacked proposal.

**Figure 1:** The standard block used in previous work and an example of double stacked block. Each block is followed by a feed-forward net similar to transformers.

To further stabilize the state space, Goel et al. (2022) used the more stable Hurwitz formulation $\boldsymbol{A}_n = \boldsymbol{\Lambda}_n - \boldsymbol{p}_n \boldsymbol{p}_n^T$, ensuring the eigenvalues of the transition matrix remain negative when $\boldsymbol{\Lambda}_n$ is negative definite. Practically, the parameters are stored in log-form: time-step $\Delta_n = \exp(z_{\Delta_n})$ and diagonal $\boldsymbol{\Lambda}_n = -\exp(\boldsymbol{z}_{\boldsymbol{\Lambda}_n})$. Regarding the initialization of this system, Gu et al. (2020) developed several options for how the transition and input matrices should be set to equip the model with long-range modeling ability which can be motivated from a memorization of signals point of view. However, all of the parameters were made trainable, discarding the initialization.

# 3 MULTI-HEAD STATE SPACE MODEL

This section introduces the novel multi-head state space model as a natural extension to the linear time-invariant structured state space model as described above. First, we discuss our configuration for the state space layer and its intialization (Section 3.1). Second, we propose novel ideas for how the architecture can be improved, including a multi-head generalization (Section 3.2). Third, a new gating mechanism is introduced which was found notably beneficial (Section 3.3). Furthermore, a bidirectional architecture is discussed (Section 3.4) and how it can be used with attention based models (Section 3.5). Finally, the compute complexity of state space models scales only linearly with sequence length allowing it to be used prior to sub-sampling layers without much added cost (Section 3.6).

## 3.1 TIED PARAMETERS & INITIALIZATION

The original design in Equation 3 describes a block diagonal simplification to a multi-variate SSM for purposes of efficiency, which can also be seen as several small independent SSMs running in parallel. However, this SSM required a smaller separate learning rate for its parameters compared to the rest of the network to achieve the best possible performance. In this work we opt for a slightly more parameter efficient blockwise isotropic implementation, similar to Gu et al. (2022b), where we restrict the copies to have shared input and transition matrices across all blocks:

$$
\begin{aligned}
\boldsymbol{A}_n &= \boldsymbol{\Lambda}_n - \boldsymbol{p}_n \boldsymbol{p}_n^T = \boldsymbol{A}_0 = \boldsymbol{\Lambda}_0 - \boldsymbol{p}_0 \boldsymbol{p}_0^T \\
\boldsymbol{B}_n &= \boldsymbol{B}_0
\end{aligned}
\qquad \forall n \in \{1, ..., D_i\}
\tag{7}
$$

but keep independent output $\boldsymbol{C}_n$ and skip $\boldsymbol{D}_n$ matrices. We also let the learning rate remain the same as the remainder of the network. By still having different time-step discretizations for each dimension of the input sequence, temporal features of different scale can be extracted while sharing the recurrence parameters. Furthermore, we choose to simply random initialize all parameters of the SSM by sampling from a the unit Gaussian distribution $\mathcal{N}(0, 1)$, apart from the diagonal matrix which is shifted $\boldsymbol{\Lambda}_0 = -\exp(\boldsymbol{z}_{\boldsymbol{\Lambda}_0}), \boldsymbol{z}_{\boldsymbol{\Lambda}_0} \sim \mathcal{N}(\ln(\texttt{scale}), 1)$. This is to ensure that the initial eigenvalues of the transition matrix are not too small and can capture enough of the input signal's history. Compared with Gu et al. (2020), the random initialization is simple but shows good performance in practice. The effectiveness of random initialization was also observed in Mehta et al. (2022). Also, it provides the potential for the multi-head state space model (see section below) in breaking the symmetry between heads instead of using the same deterministic initialization.

## 3.2 STACKED & MULTI-HEAD GENERALIZATION

The basic architecture described above is highly parameter efficient, dominated by the linear layer, and is significantly smaller than multi-head attention and feed-forward nets (FFN). Therefore, the design is expanded by stacking $\boldsymbol{y} = \boldsymbol{f}_s(\boldsymbol{f}_{s-1}(...\boldsymbol{f}_1(\boldsymbol{u})))$ each with its own set of parameters, see an example in Figure 1b. This would balance the parameters in terms of the ratio between FFN and SSM layers and give the whole model more depthwise/temporal flexibility.

Moreover, we found it useful to project the input into several different lower-dimensional representations which are passed on to independent structured state space models, in a design similar to multi-head attention, see Figure 2a. Not only does having shared recurrence parameters across a structured SSM stabilize training, but it gives each structured model in a multi-head framework the flexibility to learn both meaningful time-steps and different types of temporal dynamics. This stacked multi-head state space model (MSSM) requires an additional initial projection layer compared to its single-head implementation, leading to a small increase in parameters. In practice, the projection would take a sequence of $D_i$-dimensional features and project it down to $H \in \{2, 4, 8, \dots\}$ different, $\bar{D}_i = D_i / H$-dimensional signals, each of which is processed by isotropic blockwise diagonal SSMs. Since, the effective dimension $H \cdot \bar{D}_i = D_i$ remains the same, the cost of a multi-headed approach boils down to an extra linear projection layer.

## 3.3 INTER-HEAD GATED LINEAR UNIT

By default, prior work has used the GELU activation function. In some experiments (Gu et al., 2022a; Goel et al., 2022; Mehta et al., 2022) the GLU activation (with $C = 2$) was also found

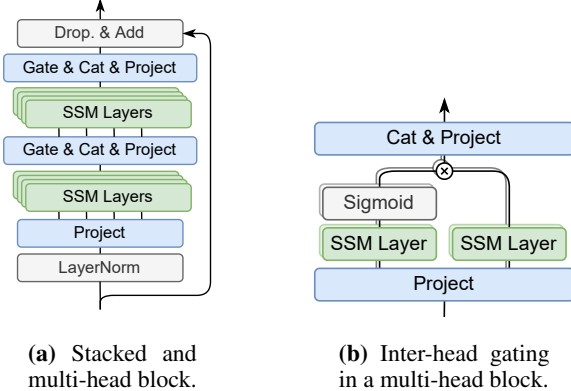

**(a)** Stacked and multi-head block.

**(b)** Inter-head gating in a multi-head block.

**Figure 2:** Multi-head generalization and novel inter-head gating idea. Instead of gating the two-dimensional output of a single SSM, this approach is based on elementwise gating of a head using the corresponding output of a different head.

beneficial which was achieved by gating the channels of an individual sub-state space model:

$$a_n = y_{n1} \cdot \sigma(y_{n2}), \ \boldsymbol{y}_n \in \mathbb{R}^2, \ \forall n \in \{1, ..., D_i\} \tag{8}$$

However, a multi-head state space architecture with $H$ headed outputs $\{\boldsymbol{y}_n^{(h)} \in \mathbb{R}^2\}_{h,n=1}^{H,\bar{D}_i}$, offers many more gating possibilities. We propose an *inter-head gating* (IHG) approach in which the output is computed by mixing the heads according to (where $\sigma$ refers to the sigmoid):

$$a_{nc}^{(h)} = y_{nc}^{(h)} \cdot \sigma(y_{nc}^{(h+H/2)}), \ \forall c = \{1, 2\} \ \ h = \{1, ..., H/2\} \tag{9}$$

allowing different heads to communicate and gate each other, generally leading to improved results. Clearly, flattening out the output in Equation 9 leads to the original $D_i$-dimensional output, and note that the number of heads in this approach has to be even, see Figure 2b. While our approach is wildly different, the concept of letting heads communicate has also been proposed in Shazeer et al. (2020) in the context of attention where linear layers mix information across heads, and was found to improve performance. However, unlike the last-mentioned, our approach allows for communication between different heads without any extra parameters.

## 3.4 BIDIRECTIONAL DESIGN

So far the proposed architecture has a unidirectional design however, for better performance, bidirectional context is needed. Therefore, we simply concatenate the output of independent forward and backward stacked multi-head modules and project it back down to the original input dimension (Cornegruta et al., 2016; Goel et al., 2022). This new module is then wrapped in a pre-norm block similar to the architectures above:

$$\begin{aligned} y &\leftarrow \mathtt{Cat}\big([\mathtt{MSSM}(x), \mathtt{Rev}(\mathtt{MSSM}(\mathtt{Rev}(x)))]\big) \\ y &\leftarrow \mathtt{Linear}(\mathtt{Act}(y)) \end{aligned} \tag{10}$$

Since there is a linear layer in the bidirectional block, the final linear layer of the MSSM modules are removed. This attention free design has many advantages over multi-head attention such as memory requirements scaling linearly with sequence length, making it computationally feasible for longer sequences.

## 3.5 STATEFORMER: STATE SPACE AUGMENTED TRANSFORMER

A pure multi-head state space model architecture is attractive due to its efficiency and ability to capture both short and long range dependencies. However, since the state space model is equivalent to a linear RNN, it is expressively more limited, and is still plagued by a linearly scaling path length. Therefore, for scenarios where one is less restricted by compute limitations, we propose a model combining the MSSM with attention by simply inserting a pre-norm (bidirectional) block prior to the self-attention unit in the transformer architecture, referred to as the *Stateformer*, see Figure 3.

**Figure 3:** Stateformer: Bidirectional state space augmented transformer encoder example. It simply has an additional block prior to the attention unit with a bidirectional MSSM as seen in Equation 10.

This type of design is inspired by the Conformer (Gulati et al., 2020) where a dedicated convolutional block is inserted to handle local relations. However, we hypothesize that the advantage of the Stateformer, is the MSSM with its variable length kernel and the multi-head attention both can handle short and long-term dependencies and can complement each other. Furthermore, the attention enables maximum path lengths of 1, resulting in a possibly more flexible model.

### 3.6 MULTI-SCALE ARCHITECTURE

In many long sequence problems, it is standard practice to sub-sample or stride the input sequence before being fed to the main model for reducing computation. For example, both Listen Attend and Spell (Chan et al., 2016) and transducer (Graves, 2012) based models for speech recognition utilize this type of sub-sampling to achieve lower frame rates and shorter sequences. In this work we investigate the multi-scale architecture, using MSSM layers prior to the striding, for sub-sampling. There are two advantages of this multi-scale design: (1) MSSMs scale linearly and can potentially handle longer sequences; (2) processing the sequence prior to sub-sampling can introduce better contextual information.

## 4 EXPERIMENTS

We investigate our proposed models on large-scale automatic speech recognition (ASR) and masked language modeling (MLM) tasks, and compare with a range of transformer-based schemes on each task. Baseline and proposed models are implemented using an extension of the Fairseq framework (Ott et al., 2019).

### 4.1 AUTOMATIC SPEECH RECOGNITION

We first evaluate the proposed models on automatic speech recognition using the widely known LibriSpeech corpus (Panayotov et al., 2015), consisting of about 960 hours of speech data sampled at 16kHz. All speech recognition models were built in the transducer framework (Graves, 2012), which has three components: encoder, predictor and joiner. We train various transducers by keeping the predictor and joiner fixed, and compare attention-based and S4 (Gu et al., 2022a) baselines, and the proposed models on the encoder.

In these experiments, we used a sliding window of 25ms with a 10ms frame shift to extract 80-dimension log Mel-filter bank features which were immediately linearly projected to a dimension of 128. Depending on the model, we used one of the following sub-samplers (SS) in order to reduce the frame rate to 40ms and increase the dimensionality to 512:

- *Conv*: 2D-convolution network with an effective stride of 4 and 512 output channels (Wang et al., 2020b; Gulati et al., 2020).
- *TR*: Time-reduction layer which stacks each 4 consecutive frames (stride = 4).
- *MS*: Multi-scale front introduces MSSM layers prior to time-reductions (Section 3.6). 2 MSSM blocks are passed on to TR(stride = 2) followed by 4 MSSM blocks and a final TR(stride = 2).

The attention-based baselines used the convolutional sub-sampler and consist of 20-36 Transformer or Conformer (Gulati et al., 2020) blocks with 512-dimensional features, 8 attention heads, and feed-forward nets with a dimension expansion of 4. Different to Gulati et al. (2020), our Conformer baseline did not use a macaron style block as it was not found useful, and had the convolutional module prior to the attenion with a kernel size of 31. Also, an S4 model was prepared, for it is

a single-head SSM baseline with careful parameter initialization; the scaled Legendre (LegS) initialization is used. The S4 baseline and our proposed models simply replace the attention block of the transformer layer, and the configuration of this layer, such as number of stacks, heads, use of inter-head gating and sub-sampler design, is investigated; $D_{sh} = 32$ is fixed for all experiments. The Stateformer uses the same parameters as the transformer combined with the best MSSM configuration. The predictor will remain fixed for all experiments and contain three 512-dimensional LSTM layers with layer-norm and dropout. The outputs of both encoder and predictor are projected to 1024 dimensions and fed into an additive joiner with a single linear layer of $|\mathcal{Y}| = 4097$ sentence-piece (Kudo & Richardson, 2018) output units.

All models used the Adam optimizer (Kingma & Ba, 2015) with a learning rate linearly warming up to the peak value in 8000 iterations, fixed until the 60th epoch and thereafter, exponentially decayed by a factor of 0.96 each epoch. A dropout value of 0.10 is used in all encoders and 0.30 in all predictors and the batch size is set based on occupying maximal GPU memory. All models were trained up to 200 epochs using 32 NVIDIA A100 GPUs. In evaluation, decoding results are generated using the beam search algorithm on the transducers without any external language model. Best checkpoints for test-sets are selected on the respective dev-sets.

| Layers | Params | Stack | WER% (↓) | | | |
| --- | --- | --- | --- | --- | --- | --- |
| | | | devclean | devother | testclean | testother |
| 16 | 56.8M | 1 | 2.57 | 7.13 | 2.79 | 6.86 |
| | 66.3M | 2 | **2.36** | **6.88** | **2.52** | 6.59 |
| 20 | 67.6M | 1 | 2.42 | 6.92 | 2.67 | **6.56** |

**Table 1:** Impact of stack depth of state space model in one layer on Librispeech.

First, the impact of SSM stack depth is examined. As reported in Table 1, the double stacked 16-layer configuration (illustrated in Figure 1b) achieves similar performance to the 20-layer non-stacking model. This demonstrates that the double stacked design is more effective and will be the default topology used in subsequent experiments.

| Model | Params | #H | IHG | SS | WER% (↓) | | | |
| --- | --- | --- | --- | --- | --- | --- | --- | --- |
| | | | | | devclean | devother | testclean | testother |
| Transformer 20L | 77.5M | 8 | – | Conv | 2.45 | 5.82 | 2.62 | 6.15 |
| | 76.7M | 8 | – | TR | 2.96 | 7.09 | 3.05 | 7.18 |
| Conformer 20L | 92.7M | 8 | – | Conv | 2.17 | 5.54 | 2.43 | 5.45 |
| S4-LegS 20L | 78.8M | – | – | Conv | 2.36 | 6.60 | 2.67 | 6.47 |
| | 78.1M | – | – | TR | 2.45 | 6.88 | 2.71 | 6.72 |
| MSSM 16L | 66.3M | 1 | ✗ | TR | 2.36 | 6.88 | 2.52 | 6.59 |
| | 74.7M | 2 | ✗ | TR | 2.33 | 6.92 | 2.58 | 6.49 |
| | 74.7M | 2 | ✓ | TR | **2.13** | 6.81 | 2.47 | 6.44 |
| | 74.7M | 4 | ✓ | TR | 2.19 | 6.38 | 2.42 | 6.25 |
| | 74.7M | 8 | ✓ | TR | 2.17 | 6.49 | 2.43 | 6.20 |
| | 79.2M | 4 | ✓ | MS | 2.14 | **6.12** | **2.39** | **5.99** |
| Stateformer 16L | 96.1M | 4 | ✓ | MS | **2.06** | **5.01** | **2.27** | **5.07** |

**Table 2:** Performance of base configuration on Librispeech. #H is the number of heads; IHG is inter-head gating (Section 3.3). SS stands for the sub-sampling scheme used in the bottom layer.

Second, in Table 2, we compare the impact of multi-head, inter-head gating and sub-sampler designs on the 16-layer MSSM models. Although directly increasing the number the number of heads from 1 to 2 shows limited improvement, by turning on inter-head gating one can observe consistent gains on multi-head models. Furthermore, additional improvement can be achieved when increasing the number of heads from 2 to 8, keeping IHG activated. Continuing, we replace the time-reduction layer with a multi-scale sub-sampler which notably improves performance. Note the second row is a transformer model with a simple time-reduction (instead of a convolutional) sub-sampler, which can be seen as a fair baseline for MSSMs without the multi-scale option. In this case, the proposed models significantly outperform the respective transformer baselines of similar size. Next we couple

attention with the best MSSM configuration to form the Stateformer (described in Section 3.5). The final row of Table 2 indicates some degree of complementarity, that further gains can be achieved when combining both of these modeling techniques.

| Model | Params | WER% ($\downarrow$) | | | |
|-------|--------|---------|---------|----------|----------|
| | | devclean | devother | testclean | testother |
| Bi-LSTM (Zhang et al., 2020a) | 130M | – | – | 3.2 | 7.8 |
| Transformer | | | | | |
|    Zhang et al. (2020a) | 139M | – | – | 2.4 | 5.6 |
|    Liu et al. (2021a) | 160M | – | – | 2.2 | 4.7 |
| ContextNet (Han et al., 2020) | 112.7M | 2.0 | 4.6 | **2.1** | 4.6 |
| Branchformer (Peng et al., 2022) | 116.2M | 2.2 | 5.5 | 2.4 | 5.5 |
| Conformer | | | | | |
|    Gulati et al. (2020) | 118.8M | **1.9** | **4.4** | **2.1** | **4.3** |
|    Zhang et al. (2020b) | – | 2.0 | 4.7 | 2.2 | 4.8 |
|    Peng et al. (2022) | 116.2M | 2.2 | 5.6 | 2.5 | 5.5 |
| *Our Baselines* | | | | | |
| Transformer 36L | 129.0M | 2.16 | 5.28 | 2.32 | 5.34 |
| Conformer 24L | 133.7M | 1.95 | 4.84 | 2.21 | 5.04 |
| *Our Proposed Models* | | | | | |
| MSSM 32L | 140.3M | 1.80 | 4.96 | 2.01 | 4.61 |
| Stateformer 25L | 139.8M | **1.76** | **4.37** | **1.91** | **4.36** |

**Table 3:** Performance of large configurations on Librispeech, compared with best results in literature.

We also scale the model size up to 140M parameters to verify the performance of a large configuration. Baselines and the proposed models in this large configuration are trained using auxiliary classifiers similar to Szegedy et al. (2015), in which intermediate encoder outputs are trained to predict frame labels every 4 layers. Table 3 compares the proposed models with large-size baselines and state-of-the-art numbers in previous works [2]. Both the 32-layer MSSM and 25-layer Stateformer outperform our large-size baselines and a range of attention-based results in literature.

## 4.2 MASKED LANGUAGE MODELING

To explore our model's effectiveness on other modalities, we consider the widely used masked language modeling task on long sequences (Devlin et al., 2018). We follow Beltagy et al. (2020) in preparing a large-scale long-document corpus, including Stories (Trinh & Le, 2018), Real-News (Zellers et al., 2019), Books (Zhu et al., 2015) and English Wikipedia. This corpus, consisting of 6.6 billion tokens, is loaded as in Beltagy et al. (2020) in terms of 4,096-token long sequences.

It is prohibitive to apply the full attention in transformers to the 4096-token sequences due to the quadratic memory complexities. Therefore, we pick a few feasible transformer variants as baselines, including sliding-window attention (Beltagy et al., 2020), Performer (Choromanski et al., 2020) and Linformer (Wang et al., 2020a). The sliding window has a window size of 256; Performer uses a random feature dimension of 256 and ReLU as kernel function; Linformer has a compression ratio of 8. The MSSM model with $D_{sh} = 32$ has 8 heads; remaining hyperparameters are shared with attention baselines.

To validate that our model works well at scale, we test the attention and MSSM layers using a 24-layer model similar in size to RoBERTa-large (Liu et al., 2019). To make a fair comparison, training stops after the same number of data batches. Following Xiong et al. (2021), baselines are trained using the standard masked language modeling objective for 100K steps with a per GPU batch size of 8 on 32 A100 GPUs [3]. The MSSM is trained for 50K steps with a per GPU batch size of 8 on 64 GPUs, for rapid development. Perplexities (PPLs) are reported on the validation set.

Table 4 summarizes the perplexity of different models. The proposed MSSM achieves performance on-par with attention baselines, but shows notably improved perplexity with further training (20K

---

[2]The Conformer performance varies significantly in prior work. We list a few recent works for comparison.

[3]The training curves of these models are nearly stable after 100k steps.

| Model | Params | PPL ($\downarrow$) |
|---|---|---|
| Performer | 359.1M | 5.58 |
| Linformer | 459.7M | 4.14 |
| Sliding window | 434.7M | **3.47** |
| MSSM | 416.1M | 3.47 |
| +more updates | | **3.22** |

**Table 4:** Performance on masked language modeling. The final row shows the performance of MSSM with additional 20k training updates.

additional updates). Although the MSSM is expected to learn information from the full sequence, the performance gap to the sliding window is smaller than expected. To better understand the behavior of the proposed model, we analyze the learned state space kernels in Section 4.3.

### 4.3 MULTI-HEAD INTRODUCES MULTI-MODAL AND MULTI-SCALE MODELING POWER

Now that we have empirically demonstrate the effectiveness of multi-head state space models, we want to answer the following research question: *what does the multi heads learn from data?* Since the state space model can be presented as a convolution (Equation 4), the shape of the kernel filters can be studied to answer this question.

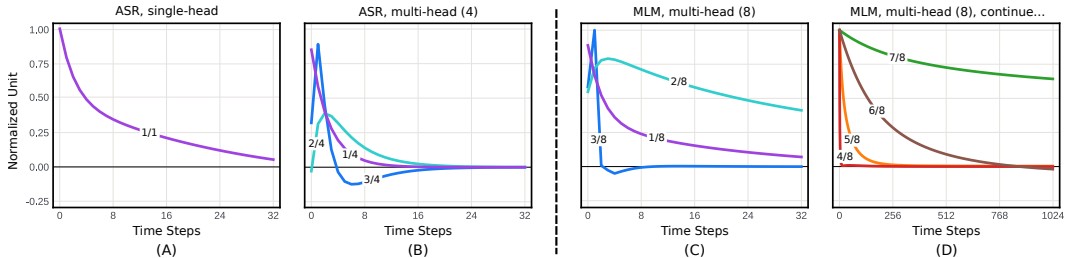

**Figure 4:** Visualization of MSSM kernels learned on LibriSpeech and masked language modeling tasks: (a): Kernel learned by single-head SSM on ASR. (b): MSSM kernels learned on ASR. (c): Kernels learned on MLM, showing multi-modal capability. (d): Kernels learned on MLM, showing multi-scale capability. (c) and (d) together are heads from the same MSSM module. To simplify the discussion, we skip plotting heads with similar patterns: 3 are plotted on ASR while 7 on MLM.

We first look into the learned multi-head kernels from ASR model (Figure 4B), and compare them with the learned single-head kernel (Figure 4A). We find the the single head kernel learned a simple monotonic decaying pattern, which computes the weighted average of history in a sliding window. On the other hand, the multi-head kernels are much more interesting — while the first head is still monotonic decaying, the kernels from the second and third heads involve context with different context concentration. In other words, multi-head state space models are able to introduce **multi-modal** modeling power.

The kernels from the MLM tasks bring additional insights. To begin with, kernels from the first three heads shows the same pattern as the ASR kernels (Figure 4C), reaffirming the multi-modal modeling power on the MLM tasks. Also, this localized behavior can be viewed similarly to that of the sliding-window attention. Additionally, kernels from the next four heads (Figure 4D) are monotonic with different decay rates, which captures history with **multi-scale** granularity.

## 5 CONCLUSION

We propose the multi-head state space model (MSSM) for sequence modeling in this paper. Parallel heads of SSMs, together with inter-head gating, are introduced, enabling multiple temporal dynamics to be effectively learned on sequence data. In addition, we combine the MSSM and attention models to form a new type of transformer architecture, referred to as Stateformer. On the LibriSpeech speech recognition task, the proposed MSSM outperforms transformer baseline by a large margin. Besides, the Stateformer further improves the performance, achieving state-of-the-art performance. On the masked language modeling task, the MSSM achieves on-par performance compared with a range of attention-based baselines trained on a large corpus.

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
