# OpenReview forum: "Multi-Head State Space Model for Sequence Modeling"
_ICLR.cc/2023/Conference — Submitted to ICLR 2023_

### Official Review · Reviewer_PQ8K · 2022-10-17

**Confidence:** 4
**Correctness:** 2
**Technical Novelty And Significance:** 2
**Empirical Novelty And Significance:** 2
**Recommendation:** 3

**Clarity, Quality, Novelty And Reproducibility:**

I do not think the paper is especially well written.  The differences between the different models is often difficult to tease apart and relate to the content introduced earlier in the paper.  The actual analysis methods for the experiments in Section 4.3 are totally missing.

I think the work is somewhat original, but the strong and immediate links to existing work are not explored, and make it very unclear to the reader what is a novel contribution.


**Strength And Weaknesses:**

# Strengths:
The paper appears to perform favourably on the tasks presented.

# Weaknesses:
The authors highlight three core contributions of the paper: multi-head parallel attention, head gating, and combination with attention through a bi-directional SSM.  I do not agree with any of these claims.

(1.1)  I am not convinced that stacking the layers is a better architecture, as the “ablations” you provide are a bit confusing (unclear how performances change with overall depth).  I also don’t understand how you allow multi-head parallel processing where S4 does not?

(1.2)  I do not believe you have correctly assessed how S4 implements attention, and so I do not believe you have “contributed” this.  (see comment below)

(1.3)  S4 uses a bidirectional SSM, which is not mentioned in your submission.  I also don’t fully understand why a transformer would _require_ bidirectionality in the SSM layer, as the transformer itself is all-to-all.  Ultimately, I find the overall architecture a little confusing.  I’d like Figure 3 to be explained a little more fully, maybe with some accompanying math close to or within the figure, to concretely define the spaces that each element operates on, its computational complexity etc.


## (2)  Connections with S4:
I find the presentation and connections to previous work, specifically in the arc of S4 papers, to be chronically lacking.

(2.1)  _I implore the authors to comment on this prior to me making my final judgement, as I may have misunderstood their claims._  Most significantly, I believe the description of S4 and the use of GLU activation is incorrect.  In Equ. (8), the authors claim that S4 uses GLU activations applied to each two-dimensional output channel per input channel.  To my understanding, this is incorrect, as S4 applies GLU to the entire $y_k^{1:H}$ vector.  The interhead gating proposed in (9) is actually a special case of the GLU activation used by S4, where the number of output dimensions is strictly tied to twice the number of input channels with size-two block-diagonal weights.  As this is the core contribution of the paper, this is a critical point.

(2.2)  The bidirectional design (Section 3.4) is also proposed by S4.  This is not mentioned.  To the reader, it appears that this bidirectional design is novel.  The parameterization used in Equ (7) is also a special case of the parameterization in Goel+ [2021], but this is not highlighted.

(2.3)  The block-diagonal interpretation was also explored by Smith+ [2022] (although I accept that this is probably acceptable as concurrent work).

(2.4)  Was S4 evaluated using the original codebase released by Gu+ [2021], or did you use your own S4 implementation?

(2.5)  Broader discussion of LSSL, S4, HTTYH, S4D, DSS, Sashimi etc is chronically lacking.

## (3) Core claim on contributions:
(This also relates to the exposition of S4 above) I do not understand the author's claim that S4 cannot capture different timescales.  The whole premise of S4 is that each SISO SSM is imbued with its own timescale, and hence can integrate information over different timescales.  Recent work, particularly How To Train Your HiPPO [Gu+, June 2022, arXiv], has explored this further, studying the kernel that S4 is implicitly using.  This paper, although recent, should still have been cited, since it tackles very similar core ideas as here.  It is also not clear to me how multi-head attention actually facilitates the claimed improvement in the ability to capture timescales.  Maybe this is true, but with the very limited intuition provided in Section 4.3, I don’t agree that better performance on benchmark tasks can be directly attributed to the new mechanism.  I am not convinced that the authors didn’t simply tune a larger model, with more parameters, flexibility, expressive non-linearities etc, until they got better performance – which then they post-hoc attribute to the capturing of timescales.

(3.1)  You state: “... long and short term dependency would both be useful in sequence modelling…”.  I do not believe you have backed the claim that yours offers this whereas S4 does not  –  this was pretty much the whole point of S4.

(3.2)  As an example, I would look at the eigen spectra of each learned latent SSM, for both SSM and the Stateformer, since this describes the temporal dynamics of the filter.

## (4)  Empirical validation:
I am not overly familiar with the particular benchmarks that the authors present.  I would like to know why they didn’t use the LRA benchmark [Tay+, 2021] or the Speech Commands dataset [Warden, 2018], as these seem like much more standard and widespread benchmarks.  These also don’t require numerous A100’s, and so also help reproducibility.

(4.1)  How were hyperparameters selected for the different methods?

(4.2)  Table 1 is very unclear to me.  Does the 16L-2S model actually have 32 layers?  Or are 16 layers arranged into eight blocks of two.

(4.3)  It is unclear what each variant in the table is exploring.  It appears to me that a soup of different layers and architectures have been tested and the best one is being reported as a substantial scientific contribution.

(4.4)  The evaluation in Section 4.3 is not explained very well.  I do not understand how the curves in Figure 4 were generated, and so I cannot evaluate the validity of the claims.  I am not even entirely sure what the curves are showing.



# Minor Weakness / Typographical Errors:
(a) Figures and tables should be floated to the top or bottom of pages.

(b) Footnotes should be avoided if possible.

(c) States are missing a time index, e.g. in Equ. (8) it should be $y_{k, n1}$, where $k$ is the time index in the sequence.

(d) I think that the claim in Section 3.2 that the shared recurrence parameters stabilise training is unsupported.  I also don’t think the second half of that claim (that multi-head frameworks give it the flexibility to learn multiple timescales) is fully supported either.  This may be me misunderstanding the analysis presented in Section 4.3.


**Summary Of The Paper:**

This paper combines continuous time state-space models with transformer themes.  The method is applied to large language and speech modelling tasks, and is compared to some baselines.  The method appears to perform well.  Some analysis of the temporal characteristics of the learned model are explored.  As far as I can tell, no code or supplementary materials were submitted.

**Summary Of The Review:**

Overall I do not think this paper is currently at the requisite level for publication.  Obvious links to existing work are missing.  I am not confident the core contribution itself is actually novel.  Standard evaluation benchmarks are missing.  The clarity of the written communication is poor.

I am willing to entertain the idea that my understanding of S4, or my understanding of particularly Equ. (8, 9) are incorrect.  If this is the case (and it is indeed how the authors claim it to be) then this would improve the paper slightly in my eyes, but probably not enough for me to rate the paper as acceptance worthy.

I also note that this paper is one of as many as half a dozen papers submitted to ICLR advancing S4.

---

> ### Author Response · Authors · 2022-11-18
> **Response to Reviewer PQ8K (Part 1/4)**
>
> Dear reviewer, we thank you for the amount of feedback received but would like to raise a number of clarifications. Some of the points raised in this review overlap with **Reviewer qqzq** and so some questions might be answered by reading the response above. However, we will answer the claims of this review in as much detail as possible.
>
>
> **Weaknesses:**
>
> We first start by responding to the points raised in this section:
>
> (1.1) We define a layer in a similar fashion to how the transformer does. Denoting Equation 10 as BiMSSM where each MSSM follows the structure in Figure 1b/2a (whether or not we use a multi-head configuration) we have: A single layer has the form: Residual(BiMSSM(x)) → Residual(FFN(x)), this is simplified for the sake of clarity. For a single head, single stack system we use MSSM(x) = Linear(Activation(SSM(x))) and for a double stack system we use MSSM(x) = Linear(Activation(SSM(Linear(Activation(SSM(x)))))), all experiments in Table 2 and onwards use the double stack architecture. Side note: The S4 does allow for parallelized processing but the MSSM is analogous to how Multi-Head Attention generalizes attention. More details can be found below.
>
> (1.2) There is some confusion regarding the statement “I do not believe you have correctly assessed how S4 implements attention” as the S4 architecture does not have any attention layers.
>
> (1.3) As mentioned in the response to  **Reviewer qqzq:** We do not claim that the bidirectional design is a novel contribution. We simply highlight that the bidirectional module is a concatenation of a forward and a backward stacked multi-headed SSMs. The idea of bidirectional sequence models has been thoroughly investigated in the context of recurrent models such as S4, RNNs, GRUs and LSTMs. For example, “Cornegruta et al 2016“ introduce a BiLSTM by concatenating independent forward and backward direction LSTMs, which is exactly how the Sashimi paper ”Goel et al 2022“ claims they have implemented bidirectionality, simply replacing the LSTM with an S4, followed by a projection down to the original dimension. Many established Deep Learning frameworks also already have this functionality. The purpose of Section 3.4 is to simply describes how we decided to design the layer. Furthermore, the Stateformer uses bidirectional MSSM because it performs better. Simply because attention is all-to-all does not mean no additional benefit could be gained by bidirectionality.

---

> ### Author Response · Authors · 2022-11-18
> **Response to Reviewer PQ8K (Part 2/4)**
>
> **Connections with S4:**
>
> (2.1) There seems to be a critical misunderstanding behind the idea of “inter-head gating”. We do agree that S4 applies GLU to the entire $y_k^{1:H}$ vector, and it does so by doubling the number of output channels (https://github.com/HazyResearch/state-spaces/blob/main/src/models/s4/s4.py#L1445). When gating, it splits the number of output channels in half, using one half to gate the other (https://github.com/HazyResearch/state-spaces/blob/main/src/models/s4/s4.py#L1525). Equation (8) given in our paper, which describes the S4 approach, is a simple example of this where the number of output channels is set to 2. Note that since the gating mechanism is a pointwise operation, we drop the time-index out of simplicity.
>
>
> Original S4 Gating: Given an input signal of batch B, length L and dimension H, which is processed by an SSM will result in a tensor of shape (B, C, H, L), where C is the number of output channels. The gating is implemented as follows:
>
> `y, yh = rearrange(y, 'b (s c) h l -> s b c h l', s=2)`
>
> `y = sigmoid(yh) * y`
>
> Which basically means that half the number of output channels are used to gate the other half.
>
> Our approach is different in the following way: Given an input signal which has been projected down to Q different heads, each of which with a dimension of H, collected in a tensor of size (B, Q, H, L), processed by Q different SSMs will result in a tensor of size (B, Q, C, H, L), with C output channels.
>
> Our inter-head gating idea, while simple, does the following (this is simple pseudocode to highlight the difference):
>
> `y, yh = rearrange(y, 'b (s q) c h l -> s b q c h l', s=2)`
>
> `y = sigmoid(yh) * y`
>
> By splitting the heads in half, as opposed to output channels, it allows different heads to gate each other and communicate. The implementation is simple and the idea can understandably be conflated with the original S4 design but is novel and allows the multi-head SSM to achieve better performance. For a full picture, when ‘inter-head gating’ is **deactivated** we use the original S4 design:
>
> `y, yh = rearrange(y, 'b q (s c) h l -> s b q c h l', s=2)`
>
> `y = sigmoid(yh) * y`
>
> but this achieved worse results in our experiments, see Table 2.
>
> (2.2) Regarding bidirectionality, see the response in (1.3). Regarding Equation 7 we have already stated in Section 2.2: "To further stabilize the state space, Goel et al. (2022) used the more stable Hurwitz formulation $A_n = \Lambda_n − p_np^T_n$"
>
> (2.3) We were unaware that “Smith et al 2022“ covered a block-diagonal view. A reference can be included.
>
> (2.4) The S4-LegS 20L in Table 2 was implemented using the S4 code provided by https://github.com/HazyResearch/state-spaces/. Practically, in addition of trying the bidirectional flag provided by the code, we also re-implemented the bidirectional module by naive concatenation of a forward and backward S4 which is projected back down to the original dimension. The latter achieved better performance and was reported in Table 2.
>
> (2.5) Much of the work in LSSL, S4, HTTYH, S4D, DSS surrounds specific initialization of the SSM. While this work could be covered it is also important to raise a note in GSS (“Long Range Language Modeling via Gated State Spaces”, Mehta et al 2022) which states that “The effectiveness of random initialization is in contrast to the findings of Gu et al. [2022a] and Gupta et al. [2022]. [...] We do however note that the experiments in our setting of large-scale language modeling are conducted on orders of magnitude larger scale of compute than what is used in the tasks considered in these works.”

---

> ### Author Response · Authors · 2022-11-18
> **Response to Reviewer PQ8K (Part 3/4)**
>
> **Core claim on contributions:**
>
> To clarify, we never claimed that S4 cannot model different time-scales. We simply point out that our multi-head architecture allows each head to derive different dynamics while each time-step controls the scale of dynamics. In Section 4.3 we explore this aspect in detail by plotting the kernels (Equation 4) themselves as these are the filters that we convolve with the input signal directly. It takes into account both the type of dynamics that is captured by (A, B, C) but also the impact of the time-scale. Figure 4A shows the kernel of a single head layer while 4B shows how each head has its own kernel dynamics. The latter was generated using ‘inter-head gating’ and so this possibly shows how different heads gating each other might have different kernel dynamics, but this needs more analysis. Regarding the statement “I am not convinced that the authors didn’t simply tune a larger model, with more parameters, flexibility, expressive non-linearities etc, until they got better performance” we argue we have slowly built up each contribution in Table 1 & 2 in a fairly parameter matched setting, and have backed each idea by performing large scale experiments on LibriSpeech. It is not straightforward to achieve (near) SOTA performance a large scale established benchmark unlike synthetic tasks (see ‘*Empirical validation*’ below).
>
> (3.1) We do not claim that multi-head SSMs are designed to better capture different time-scales, but instead, that it is designed to capture different style of dynamics and time-scales jointly, and that this is aided by a random initialization.
>
> We do state that the S4-LegS for example is initialized in a homogenous manner, by letting each sub-state space use the exact same HiPPo initialization for all pairs of (A, B), and that this could have a limiting impact on the diversity of dynamics it could model. Nevertheless, we will refrain from making sweeping statements without backing it with some analysis.
>
> (3.2) While the eigenpairs of the transition matrix could be an interesting avenue to investigate, we chose to directly plot the kernel itself (Equation 4) as this takes into account the interaction between time-steps and (A, B, C).

---

> ### Author Response · Authors · 2022-11-18
> **Response to Reviewer PQ8K (Part 4/4)**
>
> **Empirical validation:**
>
> We strongly claim that LibriSpeech is a much more established benchmark than either LRA or Speech Commands. First of all, LibriSpeech is the de facto standard benchmark in Automatic Speech Recognition and has been used since 2015 to measure advances in the speech community. Secondly, LRA consists of synthetic tasks (which they mention in the paper themselves) and are not the best indicators for real world performance on real tasks. Thirdly, the LRA repository ‘https://github.com/google-research/long-range-arena’ clearly states in the ‘Apples-to Apples setting’ that for a fair comparison, new proposed models have to follow similar architecture. Many recent works (including S4) have not followed this recipe making a fair comparison between models conceivably much harder, conflating gains from the proposed model, architecture modifications and optimization changes. Furthermore, while S4 for example obliterates all transformer style approaches on LRA & Speech Commands it is unable to outperform the transformer on the standard large-scale language modelling, WikiText-103 task. We do acknowledge the significance of S4 being able to achieve significant speedups but we stand firm that progress on LRA and Speech Commands is not a strong enough indicator of how good a model is in diverse, multi-modal, real tasks.
>
> (4.1) Many of the hyper-parameters were fixed for all models apart from batch-size and learning-rate. Batch-size was chosen as to maximise GPU memory utilization and learning rate was set to 0.001 for all attention based models and to one of {0.004, 0.006} for S4/MSSM models. We run experiments over different number of layers, stacks, number of heads, the use of inter-head gating and finally the multi-scale front-end for speech recognition. We have shown in Table 1 & 2 the results of this search and simply use the best architecture for the large models in Table 3.
>
> (4.2) See point (1.1) for details. The number of stacks determines the architecture of the BiLSTM(x) and all models with L layers will have the form L x {Residual(BiMSSM(x)) → Residual(FFN(x))}. This could be clarified in the paper.
>
> (4.3) In Table 1 we explore the effect of ‘going deeper’ ‘16L-1S’ → ‘20L-1S’ vs stacking ‘16L-1S’ → ‘16L-2S’ in a parameter matched setting. We find that the 16L-2S configuration is slightly smaller and better performing. In Table 2 we further explore this 16L-2S configuration by introducing multi-head, IHG and MS where it shows that multi-head combined with IHG can achieve lower WER. The MS option also further decreases WER outperforming Conv + Transformer in all but dev-other.
>
> (4.4) In Section 4.3 we state “Since the state space model can be presented as a convolution (Equation 4), the shape of the kernel filters can be studied to answer this question”. The plots in Figure 4 simply show the kernel itself.
>
>
> **Minor Weakness / Typographical Errors:**
>
> Thank you for finding minor errors, these will be fixed. Time-index was dropped in Equation 8, 9 since gating is a pointwise operation and would overload the equations. Regarding point (d) we will include additional results analysing the impact of sharing parameters. Finally, the statement made was that we can learn meaningful time-scales and kernel dynamics jointly.

---

> ### Comment · Reviewer_PQ8K · 2022-11-22
> **Response**
>
> Thank you to the authors for their detailed response.  Unfortunately, I am not swayed by any of the responses.
>
> Foremost, I strongly believe the authors are actively trying to downplay the clear links to existing work, e.g. Section 3.4 where the citation to Sashimi is slipped in in parenthesis, from which the authors have clearly taken this idea from (even if it does pre-date even Sashimi in the wider literature).  To a novice reader, this paper sounds fantastic, but it is really a number (maybe as small as two) of small architectural tweaks on the foundational work:  evidenced by being encapsulated in a simple change to the einops operation and stacking some layers inside a residual layer (for which, maybe permissibly, a hat-tipping citation to ResNet is also missing).  I also believe this is why the authors didn't apply their method to the now-absolutely standard LRA tasks.  I wish the authors had angled this as an empirical contribution.  The architectural tweaks seem to help on some interesting tasks with broad appeal.
>
> The paper is also still too cluttered and unclear in too many places for me to be confident that the authors could bring it up to conference-publication standard for a camera ready version.
>
> Ultimately, the submission alone is not of publication-quality, and furthermore, also misrepresents existing work.  Unfortunately, I believe the latter is absolutely terminal, irrespective of other points.  I implore the authors to overhaul the presentation, and pay due heed to existing work in future revisions, instead of trying to skirt around it.
>
> PQ8K

---

> > ### Author Response · Authors · 2022-11-24
> > **Response to Reviewer PQ8K Round 2 (Part 1/2)**
> >
> > To **Reviewer PQ8K**, we find that your response in many cases completely disregards many of the points we have raised. Therefore, our response will constitute of two parts:
> >
> > 1. We will respond to your rebuttal ‘response’
> > 2. Review some of the points we made in our first rebuttal that were disregarded by you.
> >
> >
> > **Part 1:**
> >
> > 1. The first claim made by reviewer PQ8K is that we are ‘actively trying to downplay the clear links to existing work’ giving the example of **Section 3.4 Bidirectional Design** where we have cited the Sashimi ‘in parenthesis’. The fact that the criticism against the paper partly comes down to citing a paper in parenthesis, which is standard in academia, shows that the review is lacking in quality and constructive feedback/criticism. Furthermore, the SaShiMi paper, i.e. Goel et al 2022, is **cited 4 times in this paper (Section 2.3, 3.3, 3.4)**.
> >     1. While we haven’t expected you to read the response to other reviewers, we believe **Response to Reviewer qqzq (Part 2/2)** would make further clarifications regarding the problems with the bidirectional approach in Sashimi. However, to reiterate the main point: The bidirectional approach as described in Sashimi **is not** how it was implemented in practice, and the results in **General Response to all Reviewers** shows the discrepancy in performance.
> >     2. Therefore, while we have cited Sashimi in **Section 3.4**, the Sashimi paper is still problematic, and this was confirmed by **Reviewer qqzq**: “**Bidirectional Implementation** You're right; the S4 code [which is also the same codebase Sashimi uses] does implement it different than Sashimi describes, and I agree that this particular formulation was not novel from Sashimi and has been used in some form for a long time.”
> >     3. We will furthermore, scale up the bidirectional = True models to match the number of parameters (~78M) but the difference in performance 9.4% vs 6.5% is too large to be accounted for by the smaller number of parameters.
> > 2. Characterising our contributions as ‘small architectural tweaks’ is a personal choice. The fact that a contribution such as *inter-head gating* is simple to implement could be argued is an **advantage** not a **drawback**.
> > 3. Stating that we are ‘actively trying to downplay the clear links to existing work’ and is the reason why the reviewer believes ‘the authors didn't apply their method to the now-absolutely standard LRA tasks’ is an enormous jump in conclusion, especially when we justified our view of why LibriSpeech is a much stronger benchmark than LRA.
> > 4. Instead of stating that the paper is ‘cluttered and unclear in too many places’ and ‘misrepresents existing work’ it would be orders of magnitude much more appreciated if proper constructive feedback was given. Making these types of loaded statements is not helpful for anyone in the rebuttal process.

---

> > ### Author Response · Authors · 2022-11-24
> > **Response to Reviewer PQ8K Round 2 (Part 2/2)**
> >
> > We wrote a very comprehensive response to the reviewer with a wide range of clarifications and points raised. The overwhelming majority of the points remain unanswered for and instead received a simple reply: ‘Thank you to the authors for their detailed response. Unfortunately, I am not swayed by any of the responses.’.
> >
> > In this part of the response we will again highlight some of the points.
> >
> > 1. First it seems that the reviewer was confused about the architecture as highlighted in point (1.1) and (1.2). These still remain unanswered and the reviewer has not given any indication what was actually meant by point (1.2). From our point of view this seems like a possible misunderstanding of the paper.
> > 2. We assume that the lack of response to our points in (3.2) and (4.4) regarding the analysis in *Section 4.3* means the reviewer is content with our response? If not, it would again be useful to state why.
> > 3. The main and final point seems to be the disregard to our justification of why LibriSpeech is a much more reliable indicator of performance than LRA for real world performance. Instead it seems the reviewer is completely set on that LRA is ‘now-absolutely standard LRA tasks’. We raised a three main points justifying our view:
> >     1. LibriSpeech is the de facto standard benchmark in Automatic Speech Recognition and has been used for a significantly longer time than LRA. It is based on real world data unlike most of the tasks in LRA.
> >     2. LRA consists of synthetic tasks, which is mentioned in the original LRA paper by Tay et. al. 2020. Therefore, this is not the best indicator for real world performance on real tasks
> >     3. Thirdly, the LRA repository ‘https://github.com/google-research/long-range-arena’ (https://github.com/google-research/long-range-arena%E2%80%99) clearly states in the ‘Apples-to Apples setting’ that for a fair comparison, new proposed models have to follow similar architecture. Many papers on LRA does not satisfy this criteria including S4. This makes a fair comparison between models conceivably much harder, conflating gains from the proposed model, architecture modifications and optimization changes.
> >
> >
> > Finally we want to ask you, the reviewer, to reconsider your strong position against the paper. The reviewer specifically said in their first review “I implore the authors to comment on this prior to me making my final judgement, as I may have misunderstood their claims”. It seems this misunderstanding has been clarified, however, to our surprise it seems none of the clarifications in our comprehensive response seemed to have had any impact on the reviewer’s stance.

---

### Official Review · Reviewer_F93h · 2022-10-24

**Confidence:** 3
**Correctness:** 3
**Technical Novelty And Significance:** 3
**Empirical Novelty And Significance:** 2
**Recommendation:** 6

**Clarity, Quality, Novelty And Reproducibility:**

The state space model is equivalent to a linear RNN. So what happens if the same experiments are done, but with a RNN or LSTM? Is a SSM or MSSM really adding anything?

Transducer for ASR. Without external language model.

Code?

The subsampling methods for ASR all seem non-standard. A standard subsampling can for example be seen here: https://github.com/espnet/espnet/blob/4138010fb66ad27a43e8bee48a4932829a0847ae/espnet/nets/pytorch_backend/transformer/subsampling.py#L162
It even cites the Conformer for their Conv-based subsampling, but the Conformer uses a different type of subsampling. Yes, it also uses convolution for this subsampling frontend, but it's not just a single layer, and not 1D Conv but 2D Conv.

Table 2, where is the Conformer?

Stateformer and MSSM, how does it perform with Conv subsampling? Where do I see this? Or is this missing? Why is it missing? From table 2, from the Transformer experiments, it looks like Conv subsampling would be much better?

Computational cost? For training and recognition.

What happens with more number of heads for the MSSM? Table 2 suggest it might become even better? Why was this not tested?

How exactly does the final MSSM model look like, e.g. from table 2, 3 and 4? It's not totally clear. I assume, for ASR it is in all cases bidirecitonal? But probably not the LM? This is not really explained and defined well. Where is the SSM defined exactly? In Sec 2.1 I see some equations, but now what exactly is SSM(x)? It's not really clear to me.

In the Stateformer, how does this MSSM block look like exactly? Is this exactly the same as before (figure 2)?

Table 4 on LM, with the same number of updates, there does not seem to be a difference. With more updates, MSSM gets better, but what about the other models? This is missing from the table, and it cannot really be compared directly as is.


**Strength And Weaknesses:**


Strengths:

- Good state-of-the-art results with the Stateformer on speech recognition.

Weaknesses:

- The code is not released. No information is given on the code.
- No real improvement for language modeling.
- Speech recognition lacks experiments also with an external language model. It's good to also see the numbers without, but using an external language model is the more realistic setting for speech recognition.
- Subsampling seems to be non-standard, and suboptimal. See below.
- How does it compare when replacing the SSM by a RNN or LSTM? It is said that SSMs are equivalent to linear RNNs. So what exactly is the advantage of a SSM? This does not become clear. Such experiments should be done to really see the difference.


**Summary Of The Paper:**

Tasks: Automatic speech recognition (ASR, measure WER), language modeling (measure PPL)

Two new models are proposed, which are based on state space models (SSMs).

The newly proposed models are: MSSM and Stateformer. The MSSM purely uses the SSM, with residual connections and some layer norm. The Stateformer is like a Transformer, i.e. it uses multi-head attention, and it has a new block type which embeds the MSSM.

On ASR, on Librispeech, without using an external language model, new state-of-the-art results are achieved.

For language modeling, the new model performs just the same as the sliding window attention model.


**Summary Of The Review:**

It's an interesting extension of the SSM, applied to speech recognition and language modeling, and it achieves state-of-the-art (SOTA) results on Librispeech without external LM.

There are some weaknesses though. Some of them can be improved now by better clarification, but not all of them, where more experiments should be done.

Still, due to the good results for speech recognition, I think it's interesting enough.

---

> ### Author Response · Authors · 2022-11-18
> **Response to Reviewer F93h**
>
> We thank you for your review and the good number of comments raised. Regarding ‘weaknesses’ we are working on releasing a public version of the code. In the meantime, code has been uploaded in ‘supplementary material’.
>
> For Librispeech experiment with language model, while some papers on speech recognition tasks employ external language models to improve results, we believe that it's not the focus of this paper. The ASR + external language model results depend heavily on the quality and size of the external LM, as well as decoding parameters. It distracts the readers from understanding the quality of SSMs as compared to transformer-based models in the ASR model itself. To address your concern about the compatibility of external LM with the SSM-based ASR model, we include a table in the general response using a light-weight external LM, which does improve upon SSM ASR.
>
> SSMs are equivalent to Linear RNNs and have the benefit of being highly structured and parameter efficient, whilst also facilitating parallelized ‘convolution mode’ training. The (Bidirectional) GRUs & LSTMs can only be trained by processing inputs sequentially and would make training significantly more expensive and in-practical for large-scale models. More importantly, there’s been work covering RNN based transducers. We include a latest bidirectional LSTM result in Table 3.
>
> This was an error in the paper, the Conv front-end does in fact utilize 2D convolutions. Regarding the subsampling method, it was found that the MSSMs and Stateformers benefit less from having a 2D convolutional subsampler, while a transformer does more. This is expected as these former models have convolutional elements making the frontend a bit more redundant. We do however introduce a Multi-Scale ‘MS’ frontend, comparable to ‘Conv’ but based on the MSSM instead of standard 2D convolutions, which can make better use of longe-range dependencies. However, the experiments for Conv + MSSM/Stateformer can be included in the paper for added clarity if the reviewer believes it will add value. We also increased the number of heads but observed no difference: please check the updated Table 2.
>
> We will also clarify the exact MSSM architecture as reported in the ASR and Masked LM tasks in the paper. The configuration used for all MSSM modules is a bidirectional one which concatenates the output of a forward and backward stacked MSSM. Most experiments use 2 stacks. And yes, in the Stateformer, the forward and backward MSSM in the bidirectional block has the same form as in Figure 2.
>
> This work initially focuses on improving the quality. The baseline transformers and proposed models are all tuned to have similar model sizes and FLOPs. However, for both training (on GPU) and inference (on CPU) cost, the computation time of the MSSM is roughly 2 times of the transformer/conformer. We are actively working on decreasing the computation cost by optimizing the code and choosing more efficient architectures.
>
> Finally, on the LM task it was found that baseline models converged in performance earlier than the MSSM. This meant that the MSSM based LM could be trained further and achieve even lower perplexities.

---

### Official Review · Reviewer_qqzq · 2022-10-25

**Confidence:** 5
**Correctness:** 3
**Technical Novelty And Significance:** 2
**Empirical Novelty And Significance:** 3
**Recommendation:** 5

**Clarity, Quality, Novelty And Reproducibility:**

Clarity/Novelty: Despite being primarily based on the S4 model, the connection to this and other very closely related models is not clear in the presentation. Each individual contribution to the model is reasonable but not particularly novel, and could also be better described technically and positioned with respect to prior work.

Quality: Aside from the presentation, this paper has fairly strong application results, focusing on speech, as well as some interesting analysis (e.g. visualizations of the learned kernels).

Reproducibility: Code has been provided post-rebuttal, and I believe that the results will be reproducible when fully released.



**Strength And Weaknesses:**

Amendment after rebuttal:

The original review pointed out that many of the contributions in this work are equivalent to flags available in the public S4 model. The authors have since submitted supplementary code showing that while based on the public model, several of these newer modifications are implemented somewhat differently.

This points to one of the main weaknesses of the submission, that the proposed changes are each not particularly novel. The other main weakness is a lack of discussion and comparison to very closely related prior work.

The strengths of the paper lie in the empirical results, which show strong performance in a number of speech recognition benchmarks as well as language modeling.

A much more detailed review and discussion is contained in the comments, with suggestions for improvement in the experiments and presentation.


**Summary Of The Paper:**

This paper proposes some architectural changes to the S4 model and applies it to speech recognition and language modeling tasks.


**Summary Of The Review:**

This paper proposes several modifications to the S4 model and architecture to achieve state-of-the-art results in speech recognition. Weaknesses are a lack of clarity in the details of the model, and a major lack of discussion and comparison to related work.

---

> ### Author Response · Authors · 2022-11-18
> **Response to Reviewer qqzq (Part 1/2)**
>
> Dear reviewer, we appreciate the feedback but would like to raise a number of points regarding your view of our paper and the links to the S4 codebase.
>
>
> **Section 3.1: Tied Parameters & Initialization**
>
> In this section we simply state that we opt for a simpler configuration for the state space model by parameter tying A, B. We will in a revision of this paper include references to relevant prior work. However, we do want to point out that the paper mentioned by the reviewer “On the Parameterization and Initialization of Diagonal State Space Models” has two versions on arXiv. Only the second version (recently uploaded on the 5th of August 2022) mention the notion of parameter-tying and state that “This choice was made because it generally does not affect performance much”. In our experiments we found parameter tying to be useful to achieve the lowest possible WER.
>
> The DSS paper referenced by the reviewer introduces random initialization as an alternative approach that mimics various diagonal approximations of HiPPo based theory. It is specifically mentioned in this case that “random initializations of A do not perform well”. In the GSS paper by Mehta et al 2022 they however state: “The effectiveness of random initialization is in contrast to the findings of Gu et al. [2022a] and Gupta et al. [2022]. [...] We do however note that the experiments in our setting of large-scale language modeling are conducted on orders of magnitude larger scale of compute than what is used in the tasks considered in these works.”. A citation to Mehta et al 2022 will be included in a revision of the paper as this could be a supporting reason why random initialization works in our case for large scale speech recognition and language modeling. To conclude, not only is our initialization of the diagonal and low-rank matrices different, but is also able to achieve good performance.
>
>
> **Section 3.2: Stacked & Multi-head Generalization**
>
> Our multi-head generalization is inspired by multi-head attention, in which the original input sequence is projected into multiple lower-dimensional versions over which attention is operating. Similarly, in our implementation, each projection is processed by a completely independent SSM Layer. It is not the same as simply letting the n_ssm different SSM Layers (different copies of (A, B)) each process different sets of input channels of the same signal.
>
> Furthermore, in this code base https://github.com/HazyResearch/state-spaces/blob/b1f501c0f1499fa8b545df9d334d95542da65250/src/models/s4/s4.py#L1247 it is also stated that changing n_ssm could save some parameters but “doesn't affect performance [...] much”. In our experiments we found that the multi-head approach was vital to achieve good performance able to outperform the transformer in transducer based speech recognition. Especially when combined with IHG (see below).
>
> We also want to point out that this  n_ssm flag was recently added to the S4 codebase (August 5th) and claiming that this “appear[s] to be taken from [...] public codebases without attribution” is an extremely strong claim. Not only is our multi-head idea different, but is also able to achieve better performance compared to the single-head model.
>
>
> **Section 3.3: Inter-head Gated Linear Unit**
>
> There seems to be a small but crucial misunderstanding regarding this section—the difference between ‘**intra**’ and ‘**inter-head gating**’. We do not claim that gating the output channels (can be referred to as ‘**intra**-head gating’) of a single state space model is a novel idea proposed by us and do cite prior work (S4, Sashimi, GSS) as having found this technique a useful component in certain experiments. However, we do propose an ‘inter’ approach where instead of gating the output channels of an individual sub-state space, we gate across different sub-state spaces allowing for interactions between different **input channels**. We also included Eq. (9) right below Eq. (8) to specifically highlight the difference between our and the S4 approach. See the response to **Reviewer PQ8K** for a detailed answer.

---

> > ### Comment · Reviewer_qqzq · 2022-11-20
> > **Overall Presentation**
> >
> > I think that there are a number of places where the description of the method could use improvement. The current version could do a much better job of describing the relationship between the proposed method and prior work.
> > 1. I find it somewhat strange that the method is described in relation to SSMs at large, which is a very large and varied space, whereas the recent SSMs in deep learning (especially S4) that MSSM is directly based on are only mentioned in passing here and there, and rarely by name. Much of the intro is spent discussing older works which are actually not very related to the proposed method. Although the updated manuscript has added inline references to some related papers such as GSS and S5, the paper notably lacks discussion and comparison of the deep SSM line of work on which this model is based (see point 3 as well). It would benefit from a more clear description of the relation of the proposed method with related works, perhaps in a dedicated section.
> > 2. Section 2 seems nearly copied from the S4 paper without directly mentioning it. While this material may be viewed as background, the development of it in the present form for deep learning did involve many steps developed over various papers (e.g. LMU for actually using the RNN/CNN equivalence; LSSL for learnable $\Delta$ and the SSM kernel among other things; S4 for the current formulation that MSSM is based on)
> > 3. Similarly: in point (2.5) to Reviewer PQ8K, the authors state that "Much of the work in LSSL, S4, HTTYH, S4D, DSS surrounds specific initialization of the SSM." This seems like a pretty large mischaracterization of these prior works; there were a lot of steps involved in the parameterization and computation of these models that MSSM is using.
> > 4. The submitted code is evidently adapted from the public S4 code without attributions in the file. I notice that the module in the code is called "MHS4" (I presume "Multi-Head S4") instead of "MSSM", and seems to have been rebranded later. Although since this is supplementary research code the attribution is not an issue, but contributes to the overall impression that the paper is downplaying its connection to prior work.
> > 5. Overall, I'm not sure I agree that MSSM should even be considered as distinct from S4 at all, considering how (i) the most non-trivial parts of the model are the same, (ii) the proposed modifications are simple and entirely orthogonal from the main S4 part (iii) many of these modifications are simple and obvious enough that many are already implemented in the public S4 model
> >
> >
> > I think it is fair to say that this paper is an application of S4 with various architectural changes. I think that the application itself is fairly strong. However, the presentation and positioning of the method is confusing and often raises more questions than it answers. Ignoring the fact that many of the proposed improvements have either been described before or are implemented (perhaps concurrently) in public models already, the changes lack clear and direct comparisons to the way they differ from prior work.
> > - This would be reasonable if the paper was positioned as an extension and application of S4, and I think that would make for a respectable application paper focusing on the state-of-the-art results. However the way the method is consistently described seems to present itself as a larger departure from prior work than it actually is.
> > - If the paper wishes to distinguish itself from past work through the proposed architectural modifications, each of the proposed contributions could use much more careful description, discussion, and ablations. Several suggestions have been made in each of the sections above commenting on these modifications

---

> > > ### Author Response · Authors · 2022-11-24
> > > **Response to Reviewer qqzq Round 2 (Part 1/3)**
> > >
> > > To **Reviewer qqzq**, we thank you for your detailed response. We have prepared a response structured according to:
> > >
> > > 1. A discussion regarding open source code.
> > > 2. Ablation experiments.
> > > 3. Overall presentation.
> > >
> > >
> > >
> > > **Part 1:**
> > >
> > > **Open source code:** Summarising, it seems the core criticism against the MSSM is the resemblance between our proposal and recent implementations appearing in open source https://github.com/HazyResearch/state-spaces. The main issue is boiled down to whether the concept of ‘research’ is (1) simply implementing ideas in an open source codebase or whether it is a (2) much more comprehensive process in which one has to (i) implement ideas, (ii) investigate said ideas on various tasks, (iii) analyse the effect of the ideas & (iv) participate in a peer-review process and contribute to the scientific community. While providing open source code is of much benefit to the community, we conform to the latter point (2) in which the peer-review process is key.
> > >
> > > The following is therefore our stance on some of the criticisms raised in the review: It has become apparent that some related ideas have been implemented concurrently but differently (e.g. our multi-head vs `bottleneck` & `n_ssm` flags in S4 codebase). However we have shown our contributions does lead to improved performance on large-scale tasks as opposed to the S4 team which has not published any evidence for their ideas. Barring a submission from being published simply because an unpublished combination of features exists in other codebases also does not conform to the peer-review process that we all participate in. We give examples below:
> > >
> > > **Multi-head Generalization:** We are not aware of any published work where a ‘multi-head’ approach has been investigated and been shown effective, especially on large-scale tasks. While we do acknowledge that the `n_ssm` flag which was recently implemented in the open source code is related to our approach, our multi-head state space model is directly inspired by how multi-head attention (MHA) generalised its predecessor.
> > >
> > > Regarding the last point, we did perform comparisons between the tied (all A, B pairs are the same) and non-tied (all A, B pairs are different) S4-LegS and found that the tied version actually performed marginally better, see **General Response to all Reviewers**). Our multi-head is different in that it (1) introduces a linear projection similar to MHA and (2) uses a small number of heads {2, 4, 8} similar to MHA which is unlike any of the experiments that have been described by the S4 paper.
> > >
> > > **Stacked Layers:** Actually, the use of stacking and inter-head gating aims at fusing as much inter-head/dimensional information as possible, which turns to be effective. We were not aware that stacking was an option in the S4 codebase. While the setup is quite different, we initially did have some streaming/online (i.e., using unidirectional model to do prediction with limited future context) ASR performance numbers showing that stacking is notably more helpful; those numbers could be included in the *General Response to all Reviewers* if desired.
> > >
> > > **Inter-head Gated Linear Unit:** We thought that Equation (9) combined with Figure 2b would be sufficient to describe the idea but we can include in a future revision of the paper, pseudo-code explanations for the difference between Equation (8) and (9).
> > >
> > > **Bidirectional Implementation:** There is a long history of using bidirectional sequence models and Sashimi has not cited any. However, the more troubling aspect is that there is a contradiction between what Sashimi claims is done in their paper and what their code https://github.com/HazyResearch/state-spaces actually does, which they refer to themselves.
> > >
> > > **Parameter Tying**: This point is left to last as it touches upon a different topic—code attribution. Since the S4D paper does mention parameter tying (A, B) in their version of the S4 model we have included a reference. However we cannot find any mention of this in the original S4 publication https://openreview.net/pdf?id=uYLFoz1vlAC, which is why the paper is not mentioned. However, when we do release our code, proper code attribution will of course be included to the original https://github.com/HazyResearch/state-spaces.

---

> > > ### Author Response · Authors · 2022-11-24
> > > **Response to Reviewer qqzq Round 2 (Part 2/3)**
> > >
> > > **Part 2:**
> > >
> > > This part of the response will cover the suggested experiments. Many of the comments made by the reviewer aim at conducting ablation studies to isolate the effect of each contribution separately however, many contributions are mainly effective when used jointly e.g. multi-head and inter-head gating. However, we will respond to each point below:
> > >
> > > **Initialization:** We do not claim that the main differentiating factor between MSSM and S4 is the initialization. This is only one of many aspects that differ which have been mentioned repeatedly in the paper and throughout this rebuttal process.
> > >
> > > We found that random initialization to be effective for our Language Modeling and ASR experiments. Combined with the observation made by GSS that random initialization was effective we did not conduct further experiments. While investigating the effect of random/deterministic initialization schemes is outside the aim of our paper, if the reviewer believes this would increase the quality of the paper and would be willing to increase their score, we can conduct experiments comparing various Random and LegS initialization of A, B parameters for the MSSM models.
> > >
> > > Furthermore, the reason why we compare S4-LegS with randomly initialized MSSMs in Table 2 is because we simply wanted to use the generally best performing configuration from the S4 paper for the strongest possible baseline.
> > >
> > > **Multi-head Generalization:** We decided not continue ablations with more heads than $H = 8$ since we did not observe any better performance. Similarly, the non-tied S4-LegS actually shows worse performance than its tied equivalent on the LibriSpeech ASR task as shown in **General Response to all Reviewers**.
> > >
> > > We agree on the second point that including a parameter matched $H = 1$ would be helpful for Table 2. We will therefore, set off an experiment in which we increase the number of layers.
> > >
> > > **Inter-head gating:** When IHG is turned off we perform gating over channels, following the S4 approach. Since we use GLU in all our experiments the number of output channels is always set to 2.
> > >
> > > The code in the supplementary material is a simplified version and the channel (or intra-head) gating approach was removed as it was not found useful early on in our experiments.
> > >
> > > Finally we do agree with the point raised by the reviewer and will set off some experiments comparing when IHG is turned on or off for different number of heads $H \in \{2, 4, 8\}$.
> > >
> > > **Bidirectional**: We agree that the various forms of bidirectional approaches are interesting and can extend the results in *General Response to all Reviewers* to include parameter matched models, and can include ablations in an Appendix in a revision of the paper.
> > >
> > > However, we are confident that the difference in performance between our and the S4 codebase (the code which Sashimi refers to) implementation of bidirectionality which is 9.38% (Conv + Tied S4-LegS 20L + bidirectional = True) vs 6.47% (Conv + Tied S4-LegS 20L + our own bidirectional) WER on Test-Other is too large to be accounted for by increasing the model size by ~17M parameters.
> > >
> > > **General questions on experiments:** This is a typographical mistake on our end (we have fixed this). All training and evaluation scripts were based on an internal extension of the Fairseq codebase as we mentioned in the paper. All models are evaluated on the {dev, test}-{clean other} sets using the exact same ASR decoding hyper-parameters.
> > >
> > > What we meant was that we used to the official S4 codebase to implement the transducer encoder. In this case it follows exactly the same architecture as the transformer and MSSM models in which the S4-LegS module was wrapped in a pre-norm residual unit followed by a pre-norm residual FFN unit.
> > >
> > > If we understand this last part properly, the reviewer is asking if we explored whether or not stacking helped transformers, conformers and the S4 model. Evaluating the benefit of stacking or other aspects of architecture modifications for all baselines is out of the scope of this paper and we instead leave the baselines to their default (or best found) architectures as reported by their corresponding papers.
> > >
> > > **To conclude** while including more detailed ablation studies could provide additional insights into the impact of each idea and their combined effect, this does not take away from the fact that combining all of our contributions into an MSSM & Stateformer results in highly effective models that outperform the S4, Transformer and Conformer baselines on large-scale tasks.

---

> > > ### Author Response · Authors · 2022-11-24
> > > **Response to Reviewer qqzq Round 2 (Part 3/3)**
> > >
> > > **Part 3:**
> > >
> > > The structure of Section 1 is based on a more historical treatment of SSMs building up to the current work on LTI SSMs with special initialization to equip the models with longe-range modeling ability. Section 2 then goes deeper into the theory behind this class of models where we chose to focus on the more computational aspect of this line of work in the following order:
> > >
> > > 1. Continuous state space models
> > > 2. Discretization
> > > 3. Block-diagonal restriction
> > > 4. Convolution-recurrence duality
> > > 5. Fast kernel approach
> > >
> > > The theory behind special HiPPo initialization schemes was left out as it was not found necessary for our experiments and not as important as points 1-5 above. We do acknowledge that careful initialization seems to be very much needed for the small-scale long-range tasks in LRA & SC but not for large-scale Language Modeling and ASR tasks, and is therefore outside the scope of our paper. Furthermore, we can include citations to S4 for using the block-diagonal restriction and the diagonal/normal plus low-rank restriction of the transition matrix. However, remaining points have been known in the signal processing and control theory communities for decades, and the proper and accurate reference to those communities is required.
> > >
> > > For example it was mentioned by the reviewer that LMU used the RNN/CNN equivalence (which is a fancy way of saying Linear Recurrence/Convolution equivalence and is related to IIR filters) but Chilkuri & Eliasmith 2021 “Parallelizing Legendre Memory Unit Training” themselves refer to prior work in control theory for the link between LTI LMU and its kernel. Chilkuri & Eliasmith also state that they use the convolution theorem to compute the convolution in fourier space (using padded FFTs) which again has been known for a very long time.
> > >
> > > Next it was said that LSSL introduced the use of trainable $\Delta$ and for this we do acknowledge that a citation is missing and will include this in a revision of the paper. However, the SSM kernel is not a novel idea introduced by LSSL and the proper reference to signal processing & control theory is highly preferred. Furthermore, LSSL states that the convolution between the input and kernel can be computed efficiently using FFTs, but this had already been done by Chilkuri & Eliasmith. Finally we do cite S4 and Sashimi a number of times throughout the paper for their specific and relevant contributions.
> > >
> > > Regarding the work of “LSSL, S4, HTTYH, S4D, DSS”, our claim was an overstatement. However, much of the theory being built in HTTYH is specifically for the initialization of SSMs. And while DSS introduces a simpler version of S4 a core aspect of S4D is how to initialize SSMs with diagonal transition matrix. These three papers are not as relevant to our paper due to the core focus on initialization and the use of diagonal state spaces.
> > >
> > > The code as it appears in ‘supplementary material’, was not intended for public release. When we do release our code, the proper attribution will of course be included. Furthermore, it is quite petty to look into the naming scheme and claim that it has been rebranded later on. We changed the name to MHS4 when we uploaded the code to supplementary material. However, discussing aspects such as this is was beyond the purpose of a rebuttal process which is meant to focus on the paper itself.
> > >
> > > Lastly, claiming that “MSSM should [not] even be considered as distinct from S4 at all” because “the most non-trivial parts of the model are the same” is just factually incorrect as we have pointed out many times (multi-head, IHG, stacking etc.). And just because the ideas are “simple” does not take away from the work at all. The fact that we have shown such “simple” ideas work very effectively on large-scale tasks should be a strength, not a weakness. Finally, and at the risk of repeating ourselves, many of the features that are available in the public S4 codebase were recently pushed and those ideas (that allegedly overlap with ours) did not appear in any papers at the time of submission.

---

> > ### Comment · Reviewer_qqzq · 2022-11-21
> > **Comments on the proposed changes and experiments**
> >
> > **Initialization**
> > Adding a reference to GSS is useful in this context. However I don't think this has been sufficiently ablated.
> >
> > Given that the authors claim that the main difference between MSSM and S4 is in the initialization (although, to be clear, I'm not sure I agree that this difference can sufficiently differentiate MSSM from S4; see last post), it seems important to have ablations controlling for everything but the initialization. Although some S4 baselines appear in places in the experiments, they all seem to change multiple components at once (including potentially the codebase/training pipeline; see question below).
> >
> > **Multi-head Generalization**
> > Since increasing the heads does not affect the speed of the model, I wonder why the authors ablations stopped up to $H=8$ (Table 2). I think that increasing heads even to the maximum amount does not affect parameter count much either. And some S4 experiments show that more is much better, such as PathX
> >
> > This is especially important considering that the authors claimed that MSSM differs from prior work in that prior work had maximum heads (which is not even true, see earlier comment); either way, I think it would make sense to show the tradeoffs with more heads. Another minor point is that I find the distinction in input linear layer between $H=1$ and $H=2$ arbitrary (I think it would make more sense to have the same input linear layer for the $H=1$ row in Table 2, or perhaps add a row, to control for parameters/speed)
> >
> > **Inter-head gating**
> > I'm still confused about how this was compared to the baselines in the ablations (Table 2).
> > The ablation with IHG turned off has the same number of parameters, which indicates there is still some form of doubling the SSM dimension. Response to Reviewer PQ8K indicates that perhaps the "IHG off" ablation performed gating across the `channels` dimension instead. I don't see the implementation of this "IHG off" option in the submitted code.
> >
> > I think this ablation is quite intriguing and to me seems like the most non-trivial of the proposed modifications. I'm surprised that such a small axis reordering can make a difference. Some analysis or more ablations would be a very welcome addition (e.g. ablating it for a very different setting of the model with different #heads, sub-sampling scheme, initialization, etc.)
> >
> > **Bidirectional**
> > I like the additional ablations on this, but they don't appear to be controlled for the most critical point that the authors mentioned in the response: the two versions of bidirectional have large parameter and computational speed discrepancies.
> >
> > Also, I'm a little concerned with other differences in implementation and running models in different codebases, etc. (see below point)
> >
> > I think having more description of this in the paper compared to other potential implementations (1. the one in MSSM where concatenation is after the full S4 block 2. the one proposed in Sashimi where the concatenation happens after the linear S4 layer 3. the one implemented in the actual S4 model), along with ablations actually controlled for computation/parameters, would be nice-to-haves in an Appendix. As is, the Section 3.4 seems a little unnecessary because it is not really proposing anything new and there are no ablations on it; there are potentially more important discussions that the paper could use (see later comments)
> >
> > **General questions on experiments**
> > (Major point) In response to reviewer PQ8K, the authors state that "The S4-LegS 20L in Table 2 was evaluated using the S4 code provided by https://github.com/HazyResearch/state-spaces/". This seems to imply that different models were run in different codebases, with potentially different evaluation pipelines. Looking at descriptions of the other experiments and ablations in the paper and rebuttal does not make this clear either. This seems like a potentially large discrepancy in the ablations. Given that the MSSM code is modified from the S4 code, is there a reason it would be difficult to run S4 baselines in a controlled setting?
> >
> > Similarly, Table 2: What is the distinction between the top set of rows and bottom set of MSSM experiments? Are the top set re-produced numbers, or numbers reported from papers? Are they run from the same pipeline as MSSM or different ones? Why are they 20L vs 16L, where do the extra parameters come from (do the 20L baselines not have double stack)? This seems weird to not control if you are re-running all the baselines, especially considering the text says "As reported in Table 1, the double stacked 16-layer configuration (illustrated in Figure 1b) achieves similar performance to the 20-layer nonstacking model. This demonstrates that the double stacked design is more effective and will be the default topology used in subsequent experiments".

---

> > > ### Author Response · Authors · 2022-12-07
> > > **Updated Results**
> > >
> > > We have updated the **General Response to all Reviewers** with additional results. This includes parameter matches settings for the bidirectional approaches and stacking.

---

> > ### Comment · Reviewer_qqzq · 2022-11-21
> > **Response**
> >
> > To the authors,
> >
> > Thank you for the response and providing code and clarifications. I have examined the code and revised draft closely, which have cleared up many of the confusions. I will break this response into three parts:
> > 1. Response to the original review and rebuttal: comparison between the proposed method and prior work / online code
> > 2. Comments on the model contributions and experiments
> > 3. Comments on the overall presentation of the paper
> >
> > I have also updated the original review.
> >
> > ### The proposed contributions vs prior work
> >
> > **Tied Parameters**
> >
> > The initial review only mentioned one recent reference where the weight-tying of $A$ and $B$ was mentioned, but that doesn't mean it hasn't been done earlier. Examining earlier versions of the S4 code, it seems to be present from the beginning.
> > In the very first commit, there is [functionality](https://github.com/HazyResearch/state-spaces/blob/a0c724aae456468a271d5e5fc7415185c83cde71/src/models/sequence/ss/kernel.py#L68) for tying vs untying the parameters (`trainable={1,2}`) which is [on by default](https://github.com/HazyResearch/state-spaces/blob/a0c724aae456468a271d5e5fc7415185c83cde71/src/models/sequence/ss/kernel.py#L564) for $A$.
> >
> > In the V2 tag of the code (Feb 27 2022), there is an [explicit boolean option](https://github.com/HazyResearch/state-spaces/blob/2af126108991d214fd82ffc1899f6d4e31a1eda3/src/models/sequence/ss/kernel.py#L165) for tying all the $A, B$ parameters.
> >
> > No one can be sure whether your idea was based off this or not, but given that
> > 1. this option has appeared in every version of the public S4 model
> > 2. your code is clearly a modification of that code
> >
> > I think it is reasonable to say that the description of this in relation to the original S4 model should be clarified.
> >
> > **Multi-head Generalization**
> >
> > I am fairly certain that this proposal is in fact equivalent to the `n_ssm` flag. The MSSM implements the multi-head model with shape `l q c h`, where the `h` dimension is parameter-tied and `q` represents the heads. The S4 code implements it as shape `c (h q) l`, with the same interpretation. Note that an additional input projection is needed, as the paper notes, which is an option also available in the S4 model, so this full "multi-head" model seems to already be available in the public model. I agree that the new supplementary code is implemented in a slightly different way however.
> >
> > I also don't think that "the authors left a comment saying that this flag might not help, but when we used it it helped" is appropriate justification for distinguishing this proposed change. Also, it does appear to be used on some of the harder S4 experiments, e.g. PathX where the heads is actually set to the maximum (equal to the model dimension).
> >
> > **Stacked Layers**
> > I didn't comment on this in the original review; this does also appear to be an option in the public S4 model, but doesn't seem to be used in any experiments so it is a valid difference from the original model. The performance difference in Table 1 seems extremely minor, I wonder if there is a more compelling way to show if it is consistently better (this is a minor point).
> >
> > **Inter-head Gated Linear Unit**
> > After the author's detailed description involving the einops patterns to Reviewer PQ8K, this is much more clear now. I think that the description in the paper is still quite confusing, and easy to get confused as it involves subtle reordering of axis dimensions. I might suggest adding the same code explanation (that you gave to Reviewer PQ8K) to the Appendix with a forward reference.
> >
> > **Bidirectional Implementation**
> > You're right; the S4 code does implement it different than Sashimi describes, and I agree that this particular formulation was not novel from Sashimi and has been used in some form for a long time.
> >
> > **Comments on initial review**
> > I admit that the original review was perhaps too strong about these points, but I do believe it is accurate to say that many of the proposed contributions either
> > - has been implemented in the public S4 code since release
> > - is re-creatable through a combination of flags in the current S4 model (perhaps implemented concurrently)
> > - or is extremely similar to implementations in the S4 code, with only slight permutations of the axis order
> >
> > This may be an unfortunate coincidence and thus was easy to misinterpret.
> > In the next two sections I will evaluate these on their own merit.

---

> ### Author Response · Authors · 2022-11-18
> **Response to Reviewer qqzq (Part 2/2)**
>
> **Section 3.4: Bidirectional Design**
>
> We do not claim that the bidirectional design is a novel contribution. We simply highlight that the bidirectional module is a concatenation of a forward and a backward stacked multi-headed SSMs. The alternative that could have been investigated is to stack several bidirectional layers wrapped in a pre-norm unit. Obviously, bidirectional architectures that simply concatenate forward and backward directions have been studied previously in GRUs and LSTMs and can be strongly argued that it is not a novel idea introduced by the Sashimi paper either.
>
> Furthermore, we do want to highlight that the Sashimi paper states “We simply pass the input sequence through an S4 layer, and also reverse it and pass it through an independent second S4 layer.”. However, this does not seem to be how the bidirectional layer was implemented in practice:
>
> Based on the implementation given in https://github.com/HazyResearch/state-spaces/blob/b1f501c0f1499fa8b545df9d334d95542da65250/src/models/s4/s4.py#L1361 (as referred to by reviewer) it seems that the bidirectional flag bidirectional=True simply doubles the number of output channels (https://github.com/HazyResearch/state-spaces/blob/b1f501c0f1499fa8b545df9d334d95542da65250/src/models/s4/s4.py#L1448), followed by splitting the resulting kernel in half based on the output channels, reversing the second half and summing padded versions (https://github.com/HazyResearch/state-spaces/blob/b1f501c0f1499fa8b545df9d334d95542da65250/src/models/s4/s4.py#L1499). The final kernel is then convolved (using FFTs) with the input signal.
>
> This implementation is quite a parameter efficient approach, and does introduce bidirectional flow of information due to the relationship between Discrete Truncated Fourier Transforms and circular convolutions. However, since they create the second kernel from the same (A, B) parameters and different output matrices C, their approach is actually quite limiting. Proper implementation of independent forward and backward S4 followed by non-linearities and a linear layer to project the dimension back down to the original dimension would require more parameters but also performs much better. We include performance numbers on the speech recognition task, investigating the impact of style of bidirectional implementation in the general response to reviewers.

---

### Official Review · Reviewer_aXdZ · 2022-11-01

**Confidence:** 4
**Clarity, Quality, Novelty And Reproducibility:** see above.
**Correctness:** 4
**Technical Novelty And Significance:** 3
**Empirical Novelty And Significance:** 3
**Recommendation:** 6

**Strength And Weaknesses:**

**Strengths**

The paper is generally well-written.

This work is a quick application of the recently proposed Linear State-Space Layer (LSSL). The ideas of multi-head and a combination of bidirectional MSSM and self-attention seem to be novel.

As compared to existing SSM methods, this paper introduces a simpler parameter initialization manner and the multi-head structure. In the librispeech experiment, the Stateformer obtains the SOTA performance.

**Weaknesses**

"Stateformer 25L" performs close to "Gulati et al. 2020" (better only over testclean). So the real benefit taken by Stateformer could be questioned. Can the authors argue against this concern?

Training with the auxiliary classifiers is not common in ASR. As said in the paper, "Our Baselines" and "Our proposed models" in this large configuration are trained using auxiliary classifiers. It brings a further concern that the superior performance of "Stateformer 25L" may be partly due to the use of auxiliary classifiers. It is better to have an ablation study.

How the perplexities are calculated for masked language models?

Compared to the improved results in ASR, the improvement on masked language modeling seems weaker. MSSM needs 20K additional updates to perform better than Sliding window. The authors should compare different models for the same number of training iterations.

**Summary Of The Paper:**

This paper proposed a multi-head state space model (MSSM) and the Stateformer which is a combination of bidirectional MSSM and self-attention. The performance of MSSM and Stateformer are evaluated on Librispeech for ASR. Further, the effectiveness of MSSM is explored on the MLM task.

**Summary Of The Review:**

See above.
The major concern is about the experimental evaluations.

---

> ### Author Response · Authors · 2022-11-18
> **Response to Reviewer aXdZ**
>
> We thank you for your review. To address the review’s points under the ‘weaknesses’ section, we would like to point out that comparing "Stateformer 25L" and the results in "Gulati et al. 2020" is not entirely fair due to different model training and decoding setups. Despite these differences, our model performs better in "devclean", "devtest", "testclean" and is marginally worse in "testother". Furthermore, using a consistent setup, our own implementation of the "Conformer 24L" (which is also trained using auxiliary classifiers) does not achieve nearly as good performance as "Stateformer 25L". Given this fact and that many other publications have not been able to achieve as good results as "Gulati et al. 2020", we argue that the Stateformer results are valid and approaching state-of-the-art. Additionally, comparing "MSSM 32L" with "Conformer 24L" further shows how competitive our proposed models are.
>
> Furthermore, a brief history of work regarding the Conformer, "Gulati et al. 2020", a Google paper, first introduce the Conformer and report a 4.3% WER on test-other. A second Google paper "Zhang et al. 2020b" however, report a 4.8% WER showing that our achieved number of 4.36% WER is all the more significant.
>
> For the masked language modeling task the perplexities are computed from the log-likelihood on the validation set. Furthermore, the first 4 rows of Table 4 showcase the results when all the models were trained for the same number of updates. Regarding the last row of Table 4,  it was found that baseline models converged in performance earlier than the MSSM. This meant that the MSSM based LM could be trained further and achieve even lower perplexities.

---

> > ### Comment · Reviewer_aXdZ · 2022-12-01
> > **After reading the response from the authors**
> >
> > Thanks for your response.
> >
> > >Regarding the last row of Table 4, it was found that baseline models converged in performance earlier than the MSSM. This meant that the MSSM based LM could be trained further and achieve even lower perplexities.
> >
> > This may be tricky. "Converge in performance earlier" may be affected by inappropriate configuration of learning rates and etc. Thus, the advantage of MSSM in masked language modeling is questionable.
> >
> > Further, your reply does not answer my question "How the perplexities are calculated for masked language models?". In fact, for masked language models such as BERT, only the so-called pseudo log-likelihoods can be calculated, which are in fact not a very sound metric. I would suggest to test with auto-regressive language models.

---

### Author Response · Authors · 2022-11-18
**General Response to all Reviewers**

Source code has been uploaded and is available in supplementary material. Due to our internal process, we need additional time in open-sourcing complete data, code and scripts to replicate the numbers given in this paper. Furthermore, we would like to make a general statement regarding some reviewers’ concern that our work is not novel enough:

1. It is claimed that ideas appearing in our work have recently been implemented in open source work, but are unpublished. Notably, our ideas differ from how they were implemented in the open source code. We give examples of how these differ in response to **Reviewer qqzq** and **PQ8K**.

2. Furthermore it is claimed in the open source code that those ideas do not have an impact on performance. In our case our ideas of multi-head and inter-head gating do improve performance notably. We give examples in the response to **Reviewer qqzq**.

3. We perform our work on transducer-based speech recognition task using the well-established large-scale LibriSpeech dataset where state space models have not been previously used. Managing to obtain good results on a new task is novel in itself.


To address some of the worries raised by reviewers we have conducted several experiments where we compare the S4 implemented by the original codebase with our MSSM models. Since we argue that the bidirectional flag as implemented in the original code base is limiting (see Response to Reviewer qqzq), we compare their and our own way of bidirectionality in the table below:

| Model | Parameters | Test-Other %WER | %WER with Conv front-end |
| :--- | :----: | :----: | :----: |
Transformer 20L | 76.7M | 7.18 | 6.16 |
No Tying S4-LegS 20L + bidirectional = True | 62.6M | 10.81 | 9.44 |
Tied S4-LegS 20L + bidirectional = True | 61.6M | 10.66 | 9.38 |
No Tying S4-LegS 26L + bidirectional = True | 77.3M | 10.26 | 9.31 |
Tied S4-LegS 26L + bidirectional = True | 76.0M | 10.29 | 9.25 |
Tied S4-LegS 20L + our own bidirectional (reported as baseline in Table 2 in paper) | 78.1M | 6.72 | 6.47 |

Quick description of models:

1. *No Tying S4*: We use the original S4 code with a different (A, B) pair for every input channel (and with GLU gating)
2. *Tied S4*: All input channels share the same (A, B)
3. *our own bidirectional*: we use the original S4 but implement the bidirectionality by proper concatenation of the forward and backward directions followed by non-linearity and linear projection back to original dimension.


Clearly, the original S4 code base with the bidirectional flag does not perform well at all, and is only able to achieve better performance when we properly implement Equation 10. Note that the convolutional front-end introduces an additional ~0.8M parameters to the numbers above.

Meanwhile, our base MSSM configuration with 16 layers and 2 stacks achieves, as reported in the paper:
| Model | Parameters | Test-Other %WER |
| :--- | :----: | :----: |
Bidirectional MSSM 16 Layers with 2 Stacks | 66.3M | 6.59 |
with 4 Heads + IHG | 74.7M | 6.25 |
with Multi-Scale | 79.2M | 5.99 |

While the base model which is significantly smaller, it is able to rival the S4 models above, the final model is able to outperform the transformer which has a WER of 7.18%/6.16%.

We also investigate the effect of stacking extending the results in Table 1:
| Model | Parameters | Test-Other %WER |
| :--- | :----: | :----: |
BiMSSM 16 Layers with 2 Stacks | 66.3M | 6.59 |
BiMSSM 20 Layers with 1 Stacks | 67.6M | 6.56 |
BiMSSM 20 Layers with 2 Stacks | 79.5M | 6.29 |
BiMSSM 24 Layers with 1 Stacks | 78.4M | 6.60 |

The model with 2 stacks shows potential it could scale better to larger model sizes.

To address reviewer F93h concerns on external language model, we report our large configuration using a light-weight external 3-layer LSTM LM (shallow fusion scheme):

| Model | Dev-Clean | Dev-Other | Test-Clean | Test-Other |
| :--- | :----: | :----: | :----: | :----: |
MSSM | 1.80 | 4.96 | 2.01 | 4.61 |
MSSM+LM | 1.70 | 4.42 | 1.84 | 4.21 |
Stateformer | 1.76 | 4.37 | 1.91 | 4.36 |
Stateformer+LM | 1.64 | 3.91 | 1.79 | 4.08 |

---

### Decision · Program_Chairs · 2023-01-20

**Decision:**

Reject

**Justification For Why Not Higher Score:**

- My rationale for recommending rejection of this paper, despite its excellent empirical results on the Librispeech task, is the need for this work to be more precisely positioned with respect to the prior work on SSMs, most notably the S4 model. This rationale is summarized above in "Weaknesses."

- Much of the at times vitriolic discussion between the authors and reviewers qqzq and PQ8K revolved around the question of how novel different modeling ideas in this paper were, or whether, as the two reviewers argued, they were already implemented in the publicly available S4 code base. The issue was muddied by the fact that the S4 code base is evolving, so in several cases the design ideas in question are likely to be concurrent innovations; by the fact that some of the ideas in question were not mentioned in the original S4 paper, even if they may have been used in some of the experiments; and by the fact that the Sashimi paper did not accurately describe its implementation of bidirectional modeling. The authors of this paper correctly argued that simply having an idea isn't enough, and that it is necessary to explore it in the context of a specific problem to gauge its value. But, in the end, I agree with a comment that Reviewer qqzq made: "...it feels kind of silly to quibble over provenance claims of some 5-line modifications to a 500-line model, and it would all have been circumvented if the model was just described appropriately and points (1) and (2) were executed properly." [Points (1) and (2) being clear description of prior work in isolation and clear description and evaluation through ablation experiments of the differences from the prior work.]

- There was also a debate between the authors and reviewers over whether or not evaluations on the long-range arena (LRA) tasks is necessary. I personally do not believe that they are strictly necessary: Librispeech is a challenging, large-scale speech task that is a strong test of any sequence model. However, the paper would be strengthened by inclusion of LRA results, especially if the authors intend to publish the work in a general machine learning venue rather than in a more speech-specific venue.

- Concerns from Reviewer F93h about wanting Librispeech results with an external LM were addressed in the discussion/rebuttal period. Reviewer aXdZ raised some concerns about the evaluation of the MSSM as a language model that should be addressed by providing more specifics on the perplexity computation in the supplementary material. There are two issues: first, the comparison of models trained for a different number of updates is not really fair or reliable, and second, masked language modeling is not necessarily compatible with a perplexity computation, so more details need to be provided on that. See https://huggingface.co/docs/transformers/perplexity for a discussion of the issue.



**Justification For Why Not Lower Score:**

N/A

**Metareview: Summary, Strengths And Weaknesses:**

# Summary
This paper builds on the S4 model, proposing a number of changes that enable the model to perform very well on the 960-hour Librispeech speech recognition task when it is used as the acoustic encoder in a neural network transducer model. These changes include stacking of multiple state space models (SSMs) within a residual block, multi-head processing, and inter-head gating. Furthermore, the paper proposes combining this version of an SSM with attention to create a model called the Stateformer, which is similar to a Conformer except that the convolutional block is replaced by an SSM stack (within a residual layer) in the basic Conformer block. The paper also shows that, thanks to their low inference-time cost, SSMs may be included in the downsampling layers that are a common element in the acoustic encoder of many end-to-end speech recognition models. The paper also provides experiments showing that the proposed SSM performs well on a long-context language modeling task.

# Strengths
- The speech recognition results are state-of-the-art, and the fact that they can be obtained with a suitably designed SSM is an important contribution. Reviewers F93h and fosY and the area chair, all of whom are experts in automatic speech recognition, agree on this point.

# Weaknesses
- The paper does not position itself correctly with respect to the S4 model. While the revised paper is much improved over the original, it is still not explicit enough about the commonalities that the proposed model has with S4, most notably the use of the core S4 linear layer as a black box. This could be clarified by the addition of language that explains which parts of S4 are used as is and which are modified and by provision of ablation experiments that illustrate the value of the modifications (e.g., the change from the HiPPO initialization to random initialization). A related problem is that the paper still mischaracterizes the temporal modeling abilities of the S4 model. The introduction in the revised paper says "...in Gu et al. (2022a), the S4 approach equips SSMs with careful options on parameter initialization for long-range modeling ability. **This design would force the model to capture similar temporal dynamics on different features.**" even though in the discussion the authors stated "...we never claimed that S4 cannot model different time-scales" (https://openreview.net/forum?id=hrRNkyyGGgx&noteId=RsDM5EEGnR). _It is significant that the two reviewers who are most deeply familiar with S4, reviewers qqzq and PQ8K, independently raised concerns about the positioning of this work with respect to S4._ I'm certain that the authors will revise this work and submit it to a different venue, and I urge them to be extremely careful to position this work correctly with respect to the prior work, since not doing so poses serious reputational risks.


**Summary Of Ac-Reviewer Meeting:**

N/A